# Learning Physical Graph Representations from Visual Scenes

**Daniel M. Bear**[1,3,†], **Chaofei Fan**[1,2], **Damian Mrowca**[2], **Yunzhu Li**[4], **Seth Alter**[5], **Aran Nayebi**[6],
**Jeremy Schwartz**[5], **Li Fei-Fei**[2,1], **Jiajun Wu**[2], **Joshua B. Tenenbaum**[5,4], and **Daniel L.K. Yamins**[1,2,3]

[1]Department of Psychology, Stanford University
[2]Department of Computer Science, Stanford University
[3]Wu Tsai Neurosciences Institute, Stanford University
[4]MIT CSAIL
[5]MIT Brain and Cognitive Sciences
[6]Neurosciences Ph.D. Program, Stanford University
[†]Correspondence: dbear@stanford.edu

## Abstract

Convolutional Neural Networks (CNNs) have proved exceptional at learning representations for visual object categorization. However, CNNs do not explicitly encode objects, parts, and their physical properties, which has limited CNNs' success on tasks that require structured understanding of visual scenes. To overcome these limitations, we introduce the idea of "Physical Scene Graphs" (PSGs), which represent scenes as hierarchical graphs, with nodes in the hierarchy corresponding intuitively to object parts at different scales, and edges to physical connections between parts. Bound to each node is a vector of latent attributes that intuitively represent object properties such as surface shape and texture. We also describe PSGNet, a network architecture that learns to extract PSGs by reconstructing scenes through a PSG-structured bottleneck. PSGNet augments standard CNNs by including: recurrent feedback connections to combine low and high-level image information; graph pooling and vectorization operations that convert spatially-uniform feature maps into object-centric graph structures; and perceptual grouping principles to encourage the identification of meaningful scene elements. We show that PSGNet outperforms alternative self-supervised scene representation algorithms at scene segmentation tasks, especially on complex real-world images, and generalizes well to unseen object types and scene arrangements. PSGNet is also able learn from physical motion, enhancing scene estimates even for static images. We present a series of ablation studies illustrating the importance of each component of the PSGNet architecture, analyses showing that learned latent attributes capture intuitive scene properties, and illustrate the use of PSGs for compositional scene inference.

## 1 Introduction

To make sense of their visual environment, intelligent agents must construct an internal representation of the complex visual scenes in which they operate. Recently, one class of computer vision algorithms – Convolutional Neural Networks (CNNs) – has shown an impressive ability to extract useful categorical information from visual scenes. However, human perception (and the aim of computer vision) is not only about image classification. Humans also group scenes into object-centric representations in which information about objects, their constituent parts, positions, poses, 3D geometric and physical material properties, and their relationships to other objects, are explicitly available. Such object-centric, geometrically-rich representations natively build in key cognitive concepts such as object permanence, and naturally support high-level visual planning and inference tasks.

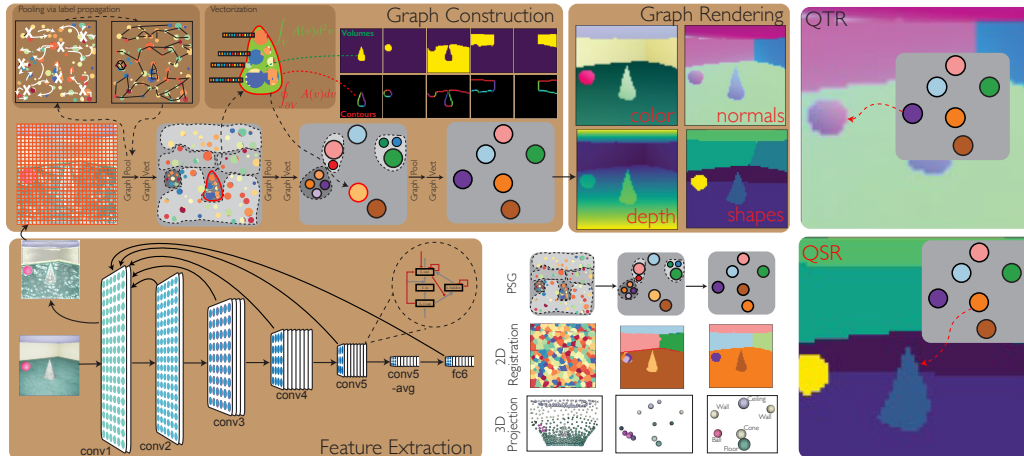

Figure 1: Overview of the PSG representation and the PSGNet architecture. Brown boxes indicate the three stages of PSGNet: (1) Feature Extraction from visual input with a ConvRNN, (2) Graph Construction from ConvRNN features, and (3) Graph Rendering for end-to-end training. Graph Construction consists of a pair of learnable modules, Graph Pooling and Graph Vectorization, that together produce a new, higher PSG level from an existing one. The former partitions existing PSG nodes into new cluster nodes, while the latter produces an attribute vector for each new node by summarizing the lower-level nodes. Three levels of an example PSG are shown (center, bottom) along with its quadratic texture (QTR) and shape (QSR) rendering (right.)

Recent work, such as MONet [4] and IODINE [17], has made initial progress in object-centric scene understanding. Learning from self-supervision, these models achieve some success at decomposing simple synthetic scenes into objects. However, these approaches impose little physical structure on their inferred scene representations and learn only from static images. As a result, they do not perform well on complex, real-world data. Recent approaches from 3D computer vision, such as 3D-RelNet [31], have attacked problems of physical understanding by combining key geometric structures (such as meshes) with more standard convolutional features. Such works achieve promising results, but require detailed ground truth supervision of scene structure.

In this work, we propose a new representation, which we call a Physical Scene Graph (PSG). The PSG concept generalizes ideas from both the MONet/IODINE and the 3D-RelNet lines of work, seeking to simultaneously handle complex object shapes and textures, to explicitly decompose scenes into their physical parts, to support the top-down inferential reasoning capacities of generative models, and to learn from real-world or realistic visual data through self-supervision, so as to not require ground truth labeling of scene components. PSGs represent scenes as hierarchical graphs, with nodes toward the top of the PSG hierarchy intuitively corresponding to larger groupings (*e.g.*, whole objects), those nearer the bottom corresponding more closely to the subparts of the object, and edges representing within-object "bonds" that hold the parts together. PSGs are spatially registered, with each node tied to a set of locations in image from which it is derived. Node attributes represent physically-meaningful properties of the object parts relevant to each level of the hierarchy, such as object position, surface normals, shape, and visual appearance.

Our key contribution is a family of self-supervised neural network architectures, PSGNets, that learn to estimate PSGs from visual inputs. PSGNets augment standard CNN architectures in several ways. To efficiently combine high- and low-level visual information during initial feature extraction, we add local recurrent and long-range feedback connections on top of a CNN backbone, producing a convolutional recurrent neural network (ConvRNN). We introduce a learnable Graph Pooling operation that transforms spatially-uniform ConvRNN feature map input into object-centric graph outputs. We also introduce a novel way to summarize the properties of spatially-extended objects using a Graph Vectorization operation. A series of alternating Pooling and Vectorization modules forms a hierarchical graph constructor. To encourage the graph constructor to produce physically meaningful scene representations, we encode key cognitive principles of perceptual grouping into the model [45], including both static and motion-based grouping primitives. To ensure that learned node attributes contain disentangled geometric and visual properties of the latent object entities, we employ novel decoders that render PSGs, top-down, into scene reconstructions or predictions.

On several datasets, PSGNets substantially outperform alternative unsupervised approaches to scene description at segmentation tasks – especially for real-world images. We also find that a PSGNet can

usefully exploit object motion both when it is explicitly available and in learning to understand static images, leading to large improvements in scene segmentation compared to an alternative motion-aware self-supervised method. The representations learned by PSGNets transfer well to objects and scenes never seen during training, suggesting that the strong constraints in the PSG structure force the system to learn general scene properties efficiently, from far less data than alternative methods. Finally, we show how the latent PSG structure identifies key geometric and object-centric properties, making it possible to compositionally "edit" the objects and attributes of inferred scenes.

**Related Work.** Unsupervised learning of scene representations is a long-standing problem in computer vision. Classic approaches study how to group pixels into patches and object instances [44, 13, 21]. Recently, machine learning researchers have developed deep generative models for unsupervised segmentation by capturing relations between scene components [10, 4, 17, 9], with varying degrees of success and generality. Beyond static images, researchers have concurrently developed models that leverage motion cues for object instance segmentation from complementary angles [16, 29], aiming to jointly model object appearance and motion, all without supervision. These models are ambitious but only work on simple, mostly synthetic, scenes. Computer vision algorithms for motion clustering and segmentation from videos achieve impressive results [55, 6], though they require high-quality annotations on object segments over time. A few papers [12] have studied unsupervised segmentation or grouping from videos, which were later used to facilitate feature learning and instance segmentation [38, 39]. However, these models do not employ an object-centric representation or model relations between scene elements. Recently, graph neural networks have shown promise at physical dynamics prediction, but they require graph-structured input or supervision [36, 32, 33, 43] – further motivating the unsupervised learning of graph representations from vision.

Researchers have developed scene representations that take object-part hierarchy and inter-object relations into account. In particular, scene graphs [30] are a representation that simultaneously captures objects and their relations within a single image. While most related works on inferring scene graph representations from visual data require fine-grained annotations, several works have studied inferring objects and their relations from videos without supervision [53, 46, 56]. These models have the most similar setup as ours, but all focus on 2D pixel-level segmentation only. Several recent papers have explored building 3D, object-centric scene representations from raw visual observations. The most notable approaches include Factor3D [51], MeshRCNN [15], and 3D-SDN [57]. 3D-RelNet [31] further extended Factor3D to model object relations. However, these 3D-aware scene representations all rely on annotations of object geometry, which limits their applicability in real scenes with novel objects whose 3D geometry is unseen during training. A few unsupervised models such as Pix2Shape [42], GQN [11], and neural rendering methods [50] have been proposed. However, none of them have yet attempted to model the hierarchy between objects and parts or the relations between multiple scene components. PSGNet builds upon and extends all these ideas to learn a hierarchical, 3D-aware representation without supervision of scene structure.

## 2   Methods

Here we give an overview of the PSG representation, components of the PSGNet architecture, and the procedure for training PSGNets. This overview explains the high-level purpose and structure of each component; formal definitions and implementation details can be found in the Supplement.

**Physical Scene Graphs.** PSGs are hierarchical graphs meant to capture the hierarchical and physical structure of scenes. Vertices in the graph, which represent objects or parts of objects, are arranged in a set of hierarchical *levels* $\{V_l \mid l = 0, 1, ..., L\}$. Edges between vertices at level $l$ and level $l + 1$ – called *child-to-parent edges* $P_l$ – intuitively represent part-whole relationships; edges between vertices at level $l$ – called *within-level edges* $E_l$ – represent abstract relationships between objects or parts. In principle, different within-level edge sets could encode different relationships (e.g. support or dynamic contact), but in this work they represent physical connections.

Beyond vertices and edges, PSGs have two additional structures: *attribute vectors* $\{A_l\}$ that label each vertex $v$ with data, meant to represent physical properties of scene elements; and *spatiotemporal registrations* (SRs) $\{S_l\}$ that explicitly link each PSG vertex in a given level to a subset of pixels in a *base tensor* $\mathcal{F}$ of shape $(T, H, W, C_0)$ (i.e. a length-$T$ movie of $(H, W, C_0)$ images. The base tensor is the set of visual features on which the PSG is built, bottom-up; it is therefore the ground level of a PSG ($l = 0$), with one vertex per spatiotemporal index $(t, h, w)$ and associated $C_0$-dimensional attribute vectors defined by $A_0(v_{thw}) \equiv \mathcal{F}[t, h, w, :]$. A registration $S_l$ partitions these indices into $|V_l|$ bins, forming a $|V_l|$-way segmentation of the base tensor (and by extension, the visual input

from which it was derived.) Moreover, the hierarchy of SRs matches the hierarchy of vertices: if a spatiotemporal position in the base tensor belongs to vertex $v$ according to $S_l$, then it also belongs to its parent vertex $P_l(v)$ according to $S_{l+1}$. Thus, the SRs indicate how elements of a scene should be hierarchically grouped into partial and whole objects. A *node* $\mathbf{n}_{l,i}$ refers to the $i_{th}$ level-$l$ PSG vertex $v_i$, its attribute vector $A_l(v_i)$, and the spatiotemporal "pixels" that have value $i$ in the registration $S_l$.

**The PSGNet Architecture.** PSGNets are neural networks that, given single or multi-frame visual input, produce a PSG representation of the scene. In particular, PSGNets create a PSG as the latent intermediate state of an encoder-decoder architecture: a PSGNet encoder first extracts features using a Convolutional Recurrent Neural Network (ConvRNN) [37], then constructs a hierarchy of PSG levels based on one layer of these features (i.e., the base tensor); finally, a set of decoders renders movie-like outputs from data encoded in the PSG 1. We describe the encoder architecture below and provide full implementation details in the Supplemental Material. We then explain how losses computed on the rendered outputs and components of the PSG intermediate are used to train PSGNets. Critically, the elements of the PSG graph structure (vertices, edges, and spatiotemporal registrations) *never* receive ground truth supervision signals, such as segmentation maps of which pixels belong to each object in a scene. Thus, PSGNets are *self-supervising graph construction algorithms*.

ConvRNN Feature Extraction. The goal of a PSGNet is to build a scene representation that reflects the physical structure of a scene: determining which parts of a visual input come from different objects, what spatial and geometric properties those objects have, and so on. For this reason, the PSG-constructing part of a PSGNet encoder needs to be based on visual features that are useful for grouping scene elements into objects and inferring physical attributes. This type of information is not easily read out from the raw pixels of RGB movies; but it is known to be decodable from the later, nonlinear layers of task-optimized convolutional neural networks (CNNs) [19]. This makes the highly nonlinear features of CNNs well-suited for some aspects of PSG construction. However, features resulting from many convolution operations, such as those near the output layer of hourglass networks, tend to be oversmoothed and have poorly defined boundaries between objects (motivating additional algorithms, such as conditional random fields, to correct these problems in the case where ground truth supervision is available, e.g. [5].) This limits the ability of CNNs to decompose scenes with sharp object boundaries (see discussion of baselines in Experiments.) We therefore want extracted features, which act as a PSG base tensor, to trade off better between representing highly nonlinear scene properties, on the one hand, and correctly segmenting pixels into objects, on the other.

To this end, we use convolutional recurrent neural networks (ConvRNNs) as PSGNet feature extractors [37]. A ConvRNN is a CNN augmented with both local recurrent cells at each layer and long-range feedback connections from higher to lower layers 1. Given an input movie, a first "pass" through the CNN backbone produces the convolutional layer activations of a standard feedforward network. Subsequent passes modulate these activations through the ConvRNN's within-layer and top-down recurrent connections, thereby combining the high-resolution features of lower layers with the highly nonlinear features of higher layers in a trainable way that can preserve the desired properties of both (see Supplemental Material for details.) In this work, we use a 5-layer ConvRNN with feedback connections from all layers to the first convolutional layer. The activations from this first layer, on the last of three passes through the ConvRNN, become the PSG base tensor.

Graph Construction. Graph construction is a hierarchical sequence of learnable *Graph Pooling* and *Graph Vectorization* operations. Graph Pooling creates within-layer edges $E_l$ between nodes from a given level, $\{\mathbf{n}_l\}$, then clusters the resulting graph to produce a new layer's vertices ($V_{l+1}$) and child-to-parent edge structure ($P_l$). This also automatically determines the new spatiotemporal registration $S_{l+1}$. Graph Vectorization aggregates and transforms the initial attributes $A_l$ according to this edge structure, yielding attribute vectors for the new nodes, $A_{l+1}$. Together, this produces the next layer of the graph, and the process repeats until a prespecified set levels have been built on top of the base tensor. The number of levels, $L$, and the particular architectures of each level's Pooling and Vectorization modules, are hyperparameters of a given PSGNet.

*Learnable Graph Pooling.* The idea behind the Graph Pooling operation is to infer which scene elements, as represented by a given level of PSG nodes, are physically connected to each other and therefore should be "perceptually grouped" into vertices of a higher PSG level. A Graph Pooling module performs this computation by predicting within-level edges $E_l$ from the input level's attribute vectors $A_l$, then clustering the resulting graph $\mathcal{G}_l \equiv (V_l, E_l)$. Each level's edges $E_l$ are predicted by a *learnable affinity function* that takes pairs of attribute vectors as input, $A_l(v) \oplus A_l(w)$, and outputs

the probability that the two vertices $v$ and $w$ are connected; these probabilities are threshholded to create a binary adjacency matrix for all vertex pairs. The architectures of different levels' affinity functions and threshholding procedures vary slightly according to the self-supervising signals that optimize their parameters, as described in PSGNet Training below and the Supplemental Material.

We cluster the graph $\mathcal{G}_l$ using the standard Label Propagation (LP) algorithm [47]. LP initializes each vertex in $V_l$ with an (arbitrary) unique segment label, then performs 10 iterations of updating each vertex with the most common label in its radius-1 neighborhood (Fig. S1). The resulting clusters (i.e., the partition of $V_l$ induced by final label assignments) are then identified as the new vertices $V_{l+1}$, with the new child-to-parent edges defined as the map from each original vertex to its cluster label. Note that this clustering process does not specify the final number of groups (unlike, e.g. $k$-means), allowing the discovery of a variable number of objects/parts as appropriate to different scenes. However, LP has the drawback of being nondifferentiable, so the parameters of Graph Pooling modules cannot be optimized end-to-end with PSG rendering losses: instead, they are trained on self-supervising *perceptual grouping losses*, described in the PSGNet Training section.

*Graph Vectorization.* What remains of building a new PSG level is to compute new attribute vectors, $A_{l+1}$. These encode physical properties of the scene elements that the newly computed vertices $V_{l+1}$ represent. Components of new attribute vectors are computed in two ways: by aggregating statistics of the lower-level attributes $A_{l+1}$ according to the predicted graph structure, and by transforming these aggregate vectors with MLPs. Computing the aggregates ensures that important information about the scene is preserved as the PSG becomes coarser at higher levels, while predicting new attributes allows PSGs to encode scene properties that are not simple functions of lower-level properties. We call this overall process "Vectorization" because it encodes information over spatiotemporally complex regions of a scene – as determined by the registration $S_{l+1}$ – in one attribute vector per vertex.

Aggregate attributes summarize the attributes of lower-level graph nodes according to the inferred perceptual groupings $P_l$ and $S_{l+1}$. Graph Vectorization modules achieve this by taking means, variances, first-order spatial moments, and higher-order statistics over each new vertex's child nodes. To retain even more information about lower graph levels, these modules also compute statistics over finer-scale portions of the segments $S_{l+1}$, including their 1D boundaries and their four spatial quadrants (Fig. 1 top left.) This is akin to a (discrete) simplicial decomposition of the segments and is described further in the Supplemental Material; that this mode of summarization preserves useful geometric information relies on Stokes' theorem for discrete simplicial complexes (see [8]).

From these aggregate attribute vectors, additional attributes are created by *graph convolution* [28] across the fully-connected graph of the $V_{l+1}$ vertices. This allows information to be exchanged, in a learnable way, between any pair of aggregate nodes (including self-pairs.) In this work, graph convolutions are implemented as MLPs $H_{l+1}^{\mathbf{new}}$ (See Supplement for details.) The final attribute vectors for PSG level $l+1$ are given by $A_{l+1}(v) = A_{l+1}^{\mathbf{agg}}(v) \oplus \frac{1}{|V_{l+1}|} \sum_{w \in V_{l+1}} H_{l+1}^{\mathbf{new}}(A_{l+1}^{\mathbf{agg}}(v), A_{l+1}^{\mathbf{agg}}(w))$.

**PSGNet Decoding and Training.** The goal of PSGNet training is to make PSG latent states represent the visual and physical structure of input scenes. In a conventional encoder-decoder architecture, the latent intermediate would be fed into a trainable decoder neural network to render outputs, which would be compared to self-generated or ground truth supervision signals [41] However, this standard approach does not encourage the latent state to reflect the physical structure of the input in an explicit or easily-decoded way: with a highly-parametrized decoder and enough training data, entangled and unstructured latent states can minimize standard training objectives, like input image reconstruction. Other approaches to unsupervised scene decomposition therefore regularize training or impose architectural constraints that aim to "disentangle" factors of variation in the training data, including the separation of scenes into discrete objects [18, 4, 17].

PSGNets not only impose constraints on the latent intermediate – by encoding scenes as hierarchical graph structures – but also render outputs with *zero-parameter decoders*, which force PSGs to represent scene components and properties much more explicitly than highly expressive decoders. PSGNet decoders effectively "paint by numbers" using predicted attribute vectors $\{A_l\}$ as the "paint" and scene segmentations $\{S_l\}$ as the regions to fill in or reconstruct. Without any trainable parameters in the rendering mechanism, PSGNets are forced to learn to encode scene properties explicitly in the PSG components themselves. We use type types of decoding, whose functional forms are detailed in the Supplement: *Quadratic Texture Rendering* (**QTR**, Fig. 1 top right), which inpaints a quadratic function of each vertex $v$'s attributes, $A_l(v)$, in that vertex's registered pixels (defined by $S_l(i, j) == v$); and *Quadratic Shape Rendering* (**QSR**, Fig. 1 bottom right), which predicts what

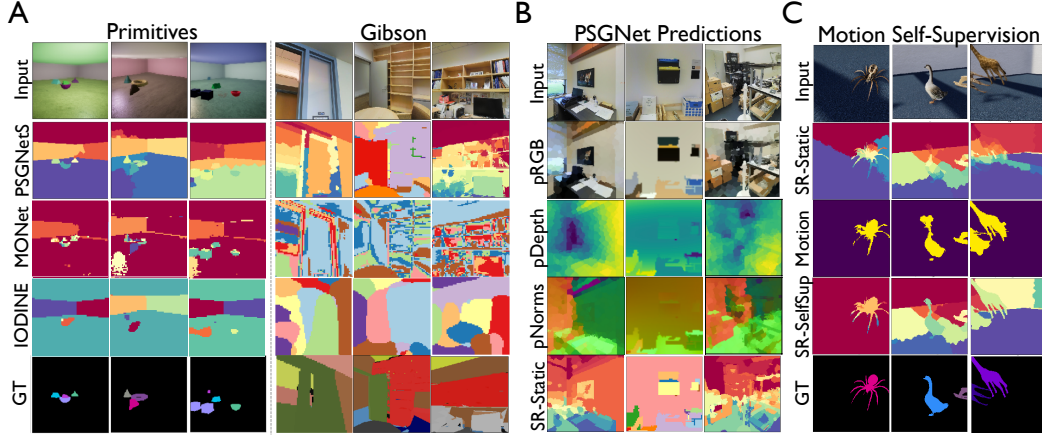

Figure 2: Visualizing the scene decompositions of PSGNets. (**A**) The predicted scene segmentations of baselines and PSGNetS. The PSGNetS segmentation is the top-level spatial registration (SR) from the PSG. (**B**) Rendering of the lowest level PSGNetS node attributes and the top-level SR on Gibson images. (**C**) Comparing the static SR (SR-Static) and the SR learned by motion-based self-supervision (SR-SelfSup) in PSGNetM. These SRs are predicted from single frames, not movies, at test time; row two (Motion) indicates where motion *would* be detected during training, according to a separate evaluation of these images as frames of movies.

2D silhouette a PSG node $v$ produces in the input scene. This elaborates a procedure developed in [7] to "draw" a shape as the intersection of quadratic signed distance function constraints, whose parameters are components of $A_l(v)$. Note that the rendered shapes do not depend directly on the registrations $S_l$, unlike in **QTR**. This means that the attributes vectors are sufficient to generate an image *via* **QSR**, a property we use below to demonstrate the symbolic structure of PSGs.

Training the Feature Extractor and Vectorization Modules with Rendering Losses. Each **QTR** and **QSR** decoder uses a distinct set of attribute vector components to render an image; the loss function applied to each image determines what properties those components will encode about the scene. In this work, we always self-supervise **QTR** outputs from all PSG levels with the RGB values and the backward temporal difference magnitudes of the PSGNet's input movie, using the standard $L^2$ loss. We also self-supervise a set of **QSR** outputs from the top PSG level on the *bottom-up scene segmentations* $S_L$, which encourages the attribute vectors $A_L$ to encode silhouette shapes as a set of parabolic boundary curves; this uses a softmax cross-entropy loss between the predicted segment index at each **QSR** pixel and the "ground truth" value $S_L[i, j]$. Finally, except where indicated, we *supervise* **QTR** renderings on actual depth and surface normal vector images provided by the training datasets, allowing PSGs to explicitly encode geometric scene information; however, depth and normal images are never given as PSGNet inputs: PSG inference requires only RGB movies. Backpropagating gradients from these rendering losses train Feature Extractor and Graph Vectorization modules, but not Graph Pooling modules due to the nondifferentiable LP operation.

Training Affinity Functions with Perceptual Grouping Principles. Each Graph Pooling module requires a loss function to optimize its learnable affinity functions. In this work we use four different types of loss function to encode four core perceptual grouping principles, which are partly inspired by human visual development [45]: *Attribute Similarity (P1):* Nodes whose attributes are especially similar (including spatiotemporal position) should be grouped together, as they are more likely to stem from the same object; *Statistical Co-occurrence (P2):* Nodes that appear often together should tend to be grouped, as this hints that they are parts of a common object. We train a Variational Autoencoder (VAE) [18] to encode attribute pairwise differences and use the reconstruction error as an inverse measure of affinity. If a node pair is common, it will be seen more often in training, so the reconstruction error should be lower than that of a rare node pair; *Motion-Driven Similarity (P3):* Nodes that move together should be grouped together, regardless of their visual appearance, as scene elements with a "common fate" likely belong to the same spatiotemporally cohesive object [45]. This principle is formalized by extending the VAE concept over multiple movie frames; *Self-supervision from Motion (P4):* Nodes within a single frame that have *previously* been seen moving together over the course of training should be grouped together. This principle trains affinity functions on *movies with object motion* so that they can make better predictions on static scenes. PSGNets are trained by choosing rendering losses and applying a perceptual grouping loss to each Graph Pooling module ($L$ modules to build an $L + 1$-level PSG), with all losses minimized at once.

Table 1: Performance of models on TDW-Primitives, TDW-Playroom, and Gibson test sets after training on each set. OP3 and PSGNetM were not trained on Gibson or Primitives as these have static test sets. Quickshift++ (Q++) receives ground truth depth and normals maps as input channels in addition to RGB.

| Models | Primitives | | | Playroom | | | Gibson | | | |
|---|---|---|---|---|---|---|---|---|---|---|
| | Recall | mIoU | BoundF | Recall | mIoU | BoundF | Recall | mIoU | BoundF | ARI |
| MONet | 0.35 | 0.40 | 0.50 | 0.28 | 0.34 | 0.46 | 0.06 | 0.12 | 0.15 | 0.27 |
| IODINE | 0.63 | 0.54 | 0.57 | 0.09 | 0.15 | 0.17 | 0.11 | 0.15 | 0.14 | 0.30 |
| Q++ (RGBDN) | 0.55 | 0.54 | 0.62 | 0.50 | 0.53 | 0.65 | 0.20 | 0.20 | 0.24 | 0.45 |
| OP3 | - | - | - | 0.24 | 0.28 | 0.31 | - | - | - | - |
| PSGNetS | **0.75** | **0.65** | **0.70** | 0.64 | 0.57 | 0.66 | **0.34** | **0.38** | **0.37** | **0.53** |
| PSGNetM | - | - | - | **0.70** | **0.62** | **0.70** | - | - | - | - |

## 3  Experiments and Analysis

**Datasets, Baselines, and Evaluation Metrics.** We compare PSGNet to recent CNN-based object discovery methods based on the quality of the self-supervised scene segmentations that they learn on three datasets. **Primitives** is a synthetic dataset of primitive shapes (e.g. spheres, cones, and cubes) rendered in a simple 3D room. **Playroom** is a synthetic dataset of objects with complex shapes and realistic textures, such as might be found in a child's playroom, rendered as movies with object motion and collisions. Images in **Primitives** and **Playroom** are generated by ThreeDWorld (TDW), a general-purpose, multi-modal simulation platform built on Unity Engine 2019. TDW is designed for real-time near-photorealistic rendering of interior and exterior environments, with advanced physics behavior that supports complex physical interactions between objects. **Gibson** is a subset of the data from the Gibson1.0 environment [3], composed of RBG-D scans of inside buildings on the Stanford campus. All three datasets provide RGB, depth, and surface normal maps for model supervision, as well as segmentation masks for evaluating predicted unsupervised segmentations. We compare PSGNet to three neural network baselines — MONet [4], IODINE [17], and OP3 [54] — and one non-learned baseline, Quickshift++ (Q++) [22], which is given ground truth RGB, depth, and normals maps as input. All three neural network baselines have similar CNN-based architectures and hyperparameters and are trained with Adam [25] as described in the Supplement.

**Static Training.** We first compared models using static images from **Primitives** and **Gibson**, with RGB reconstruction and depth and surface normal estimation as training signals. All models were given single RGB frames as input during training and testing. On **Primitives**, PSGNetS (PSGNet with static perceptual grouping only, see Supplement) substantially outperformed alternatives, detecting more foreground objects and more accurately predicting ground truth segmentation masks and boundaries than MONet [4] or IODINE [17] (Table 1). IODINE and PSGNetS also outperformed the un-learned segmentation algorithm Q++ [22]. Only PSGNetS learned to plausibly decompose real static scenes from **Gibson**. The per-segment mask and boundary detection, as well as Adjusted Rand Index (ARI) – which measures how often pixel *pairs* correctly fall into the same or distinct ground truth segments – were nearly two-fold better for PSGNetS than either learned baseline.

While PSGNetS makes characteristic grouping errors on Gibson images, its scene decompositions are broadly plausible (Fig. 2A, B.) In fact, many of its "errors" with respect to the segment labels illustrate the difference between a physical scene representation and a semantic description: what is labeled a single bookcase in the dataset, for instance, actually holds many visually and physically distinct objects that PSGNetS identifies (Fig. 2A, rightmost column.) Reconstruction of appearance and surface geometry (Fig. 2B) cannot resolve these ambiguities because objects can have arbitrarily complex shapes and textures. Instead, true "objectness" must be at least partly learned from a dynamical principle: whether a particular element can move independently of the rest of the scene.

**Motion-Based Training.** We therefore trained PSGNetM, a PSGNet equipped with the motion-based grouping principles P3 and P4, on four-frame movies from the **Playroom** dataset (see Supplement.) Tested on *static* images, PSGNetM produced more accurate scene segmentations than those of PSGNetS by learning to group visually distinct parts of single objects (Fig. 2C.) PSGNetS in turn achieved higher performance than either MONet or IODINE when trained and tested on **Playroom**. OP3, an algorithm that employs the IODINE architecture, but also learns a simple graph neural network dynamics model on the inferred latent factors to reconstruct *future* images, could in principle learn to segment static scenes better from motion observed during training [54]. While OP3 outperforms the static IODINE architecture, it does not approach PSGNet performance.

Table 2: Performance of Ablated PSGNetM variants on TDW-Playroom.

| Ablation | Base | NoLoc | NoFB | FF | UNetL | NoBo | NoVar | NoGC | NoQR | NoDN | NoAll | NoDN* |
|---|---|---|---|---|---|---|---|---|---|---|---|---|
| Recall | **0.70** | 0.59 | 0.52 | 0.47 | 0.55 | 0.63 | 0.61 | 0.57 | 0.59 | 0.41 | 0.47 | 0.65 |
| mIoU | **0.62** | 0.54 | 0.50 | 0.48 | 0.52 | 0.57 | 0.56 | 0.53 | 0.54 | 0.44 | 0.47 | 0.58 |
| BoundF | **0.70** | 0.55 | 0.50 | 0.49 | 0.53 | 0.59 | 0.58 | 0.54 | 0.55 | 0.47 | 0.46 | 0.59 |

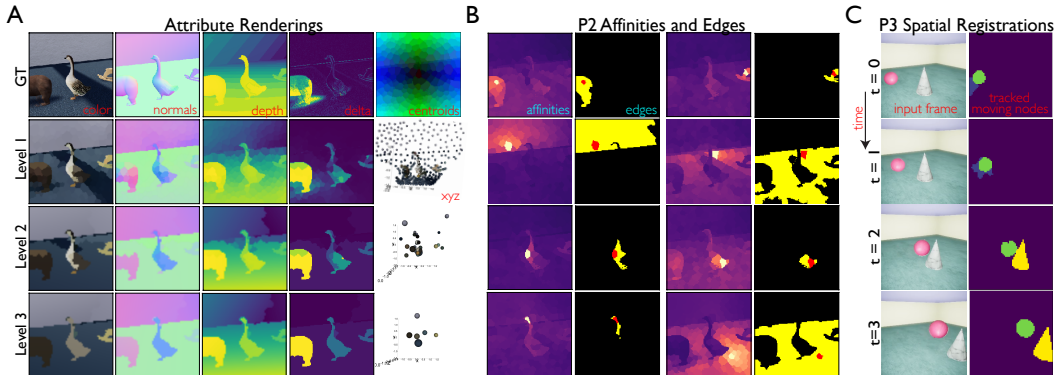

Figure 3: Visualizing the predicted components of a PSG. (**A**) The QTRs of different PSG levels for the input image in the top left. **Delta** is the magnitude of RGB change relative to the prior frame. Nodes plotted by their (x,y,z) attributes are colored by their RGB attributes, with nodes that cover <25 pixels hidden for clarity. (**B**) Eight examples of learned P2 affinities (left) and binary edges (right) between the nodes marked in yellow (left) or red (right) and other nodes. Warmer colors indicate higher affinity. (**C**) Spatiotemporal registrations inferred from motion (P3 affinity.) Segments of the same color across different frames belong to the same node.

**Learning Efficiency and Generalization Across Datasets.** Together, these results suggest that PS-GNet has better inductive biases than CNNs for unsupervised segmentation. Two further observations support this claim. First, PSGNet models are more than 100-fold more sample efficient in learning to decompose scenes. Training on either **Primitives** or **Playroom**, performance on held-out validation images saturated within two training epochs for PSGnetS and PSGNetM but only after 200-400 epochs for the CNNs on **Primitives** or 20-100 epochs on **Playroom** (though CNNs badly underfit on the latter; Table S2.) Second, PSGNet models trained on one TDW dataset largely generalized to the other – despite zero overlap between the object models in each – while the baselines did not. Test set recall on **Primitives** of **Playroom**-trained PSGNetM was 0.56 (80% of its within-distribution test performance), whereas that for **Playroom**-trained OP3 was 0.10 (42% transfer); transfers from **Primitives** to **Playroom** were 51% and 29% for PSGNetS and MONet, respectively.

To make sure PSGNet generalization was not limited to TDW scenes' appearance or geometry, we tested our models on two datasets previously used in the unsupervised scene decomposition literature, MultiDSprites (MDS) and CLEVR6 [4, 17]. Because these datasets have neither object motion nor depth and normal supervision signals, we developed a compact architecture, PSGNetS-RGB, that trained only with RGB-based self-supervision on **Playroom**. This model achieved a recall of 0.59 on the **Playroom**test set – somewhat but not dramatically lower than PSGNetS, suggesting that geometric supervision is useful but not essential for PSGNet scene segmentation. Remarkably, with no fine-tuning the pretrained PSGNetS-RGB achieved 0.75 recall on MDS and 0.70 on CLEVR6 – *higher* than on the **Playroom** test set. In sharp contrast, the best **Playroom**-trained baseline, MONet, had recall of only 0.31 on MDS and 0.05 on CLEVR6, while IODINE did not recall any objects upon transfer. The **Primitives**-trained versions of these baselines, which did not underfit and saw object shapes more comparable to those in the transfer test sets, nevertheless showed poor transfer recall (MONet: 0.5 MDS, 0.06 CLEVR6; IODINE: 0.28 MDS, 0.26 CLEVR6.) These findings suggest the PSGNet perceptual grouping inductive bias – predicting pairwise affinities between scene elements – is more efficient and more general for decomposing scenes of widely varying appearance than the convolutional inductive bias of the CNN-based models. In the Supplement we discuss these transfer experiments in more detail and describe typical failure modes for each model.

**Ablations of the PSGNetM architecture.** Deleting local recurrence (NoLoc), long-range feedback (NoFB), or both (creating a feedforward network, FF) from the PSGNetM ConvRNN increasingly hurt segmentation performance on **Playroom** (Table 2.) Moreover, a "U-Net-like" ConvRNN (UNetL) in which feedforward and upsampled recurrent features are simply summed instead of combined over

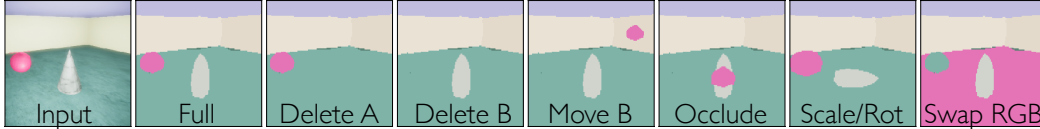

Figure 4: Symbolic manipulation of a PSG. A PSG is manually edited before shape rendering (here each shape is colored by its node's RGB attributes.) Deleting chosen nodes (DeleteA/B), moving them to new 3D locations (MoveB and Occlude), altering shape attributes (Scale/Rot), or swapping nodes' colors (Swap RGB) all change the graph rendering as predicted relative to the original (Full). These graph edits are implemented through explicit code that picks out one or more nodes and assigns new values to their attribute vectors.

time also performed worse than the base model. This is likely because the base tensor features are over-smoothed by convolution. Models without aggregation of segment border attributes (NoBo), aggregation of feature variances (NoVar), graph convolutions (NoGC), or quadratic (rather than constant) texture rendering (NoQR) also performed worse than the base model.

Removing depth and normals decoding (NoDN) without otherwise changing the PSGNetM architecture badly hurt performance; yet this model still outperformed the CNN baselines *with* depth and normals supervision (cf. Table 1), indicating that PSGNets' better segmentation does not stem only from learning explicit scene geometry. The performance of NoDN was comparable to that of a model with depth and normals training, but all ConvRNN recurrence and optional Graph Construction components ablated (NoAll). This level of performance may therefore reflect the benefit of perceptual grouping-based over CNN-based inductive biases, while the additional components mostly allow PSGNetM to learn how to integrate geometric cues into scene decomposition. Without depth and normals supervision, PSGNets perform *RGB-autoencoding*, so that too much architectural complexity and parameters may lead to overfitting. Consistent with this interpretation, a new hyperparameter search over "RGB-only" PSGNet architectures identified a relatively high-performing *feedforward* model (NoDN*) with no ConvRNN recurrence or graph convolution. This model nevertheless falls short of the base PSGNetM trained with geometric cues. Together, these results suggest that PSGNets benefit from but do not require geometric supervision. On the other hand, simpler architectures (e.g. NoAll) appear to *underfit* the combination of appearance and geometric cues, such that their performance is much worse than the hyperparameter-optimized autoencoding architecture (NoDN*).

**Visualizing learned PSGs.** Renderings from the nodes of an example PSG closely resemble the visual features they were trained to predict (Fig. 3A.) Because of quadratic rendering, a single higher-level node can represent a large, smoothly changing scene element, such as a floor sloping in depth. This often reflects underlying physical structure better than other 3D scene representations, like dense meshes [15]: a flat wall need not be encoded as jagged, and a sphere's roundness need not depend on how many surface points are supervised on. PSG edges do not have observable supervision signals to compare to. However, rendering the affinity strengths and edges from a chosen node to other nodes (Fig. 3B) shows that the strongest static (P2) relationships are within regions of similar visual or geometric attributes, as expected from their frequent co-occurrence. Regions that differ only in some attributes (such as lighted and shadowed parts of the floor) can still have high affinity – even spreading "around" occluding objects – as this common arrangement can be reconstructed well by the P2 VAE. When shown a movie, P3 affinities group nodes that are moving together. This determines the tracking of moving nodes, such that nodes in different frames are identified with a single spatiotemporally extended object (Fig. 3C.) Using these moving segments to self-supervise the P4 affinity function allows for grouping visually distinct scene elements, but also can overgroup similar-looking, nearby objects (Fig. 2C.) The Supplement contains more PSG examples.

**The symbolic structure of PSGs.** We illustrate the symbolic properties of a PSG by "editing" the nodes of its top level and observing effects on its rendering (Fig. 4.) If a PSG correctly represents a scene as discrete objects and their properties, then manual changes to the nodes should have exactly predictable effects. This is in fact what we observe. Deleting a node from the graph removes only that node's rendered shape, and changing an attribute (e.g. 3D position, shape, or color) affects only that property. The "filling in" of vacated regions indicates that a node's representation extends through visually occluded space. Because PSGs support symbolic instruction – here coded manually – future work will allow agents to *learn* how to physically manipulate scene elements to achieve their goals. Furthermore, an agent equipped with a PSGNet-like perceptual system could learn a form of *intuitive physics*: a model of which changes to the PSG are most likely. Given the promising results of applying graph neural networks to dynamics prediction problems [36, 32, 33, 43], the structured latent state of PSGNet seems well-poised to connect visual perception with physical understanding.

## Acknowledgements

This work was funded in part by the IBM-Watson AI Lab. D.M.B. is supported by an Interdisciplinary Postdoctoral Fellowship from the Wu Tsai Neurosciences Institute and is a Biogen Fellow of the Life Sciences Research Foundation. D.L.K.Y is supported by the McDonnell Foundation (Understanding Human Cognition Award Grant No. 220020469), the Simons Foundation (Collaboration on the Global Brain Grant No. 543061), the Sloan Foundation (Fellowship FG-2018- 10963), the National Science Foundation (RI 1703161 and CAREER Award 1844724), the DARPA Machine Common Sense program, and hardware donation from the NVIDIA Corporation. We thank Google (TPUv2 team) and the NVIDIA corporation for generous donation of hardware resources.

## Broader Impact

This work attempts to bring computer vision closer to its human counterpart by explicitly modeling objects, their relationships, and their physical properties – aspects of what in Psychology has been called "Intuitive Physics." Although these are merely early steps in this direction, we ultimately hope that having computer vision algorithms and agents that can interpret these aspects of our physical world will begin to address some of the well-known limitations of existing Deep Learning models: that they can be fooled by non-physical changes to their visual input and can easily overfit to statistical biases in their training data that do not reflect the true structure of the data-generating process. To this end, we have focused here on making the representations learned by our models interpretable and easily interrogated by people – indeed, we think this is one of the largest gaps between computer and human vision and the main reason that current algorithms have not yet matched their data categorization success on more physical tasks.

That said, as algorithms (including ours) become able to reason about the real, physical world in more human-like ways, we will need to further ensure that they are only deployed on real-world tasks with careful consideration and human supervision. An algorithm that is able to manipulate objects in its environments could be used to do so in either beneficial or malicious ways, much as (e.g.) scene-categorizing CNNs can be put to ever broader purposes. Moreover,*errors* in how an algorithm such as ours interprets physical scenes – for instance, mistaking an unstable multi-object structure for a single stable object – could be costly if the model were relied on for understanding or altering scenes. We do not think that, as of now, our work will put any particular group of people at advantage or disadvantage because it is still being developed on relatively limited synthetic and real datasets with inanimate objects only; but we will be cautious in extending it to more human-centric environments.

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
