[Supplementary Material]

## S1 Physical Scene Graphs

A PSG is a vector-labelled hierarchical graph whose nodes are registered to non-overlapping locations in a base spatial tensor. Formally, for any positive integer $k$ let $[k] := \{0, \ldots, k-1\}$. A *physical scene graph* of depth $L + 1$ is a hierarchical graph $\mathcal{G} = \{(V_l, A_l, E_l, P_l)|l \in [L+1]\}$ in which $V_l = [|V_l|]$ are layer $l$ vertices, $A_l : V_l \to \mathbb{R}^{C_l}$ are $C_l$-vector-valued attributes, $E_l$ is the set of (undirected) within-layer edges at layer $l$, and for $l \in [L]$, $P_l : V_l \to V_{l+1}$ is a function defining child-parent edges. We also require that for some tensor $\mathcal{F} \in \mathbb{R}^{T \otimes H \otimes W \otimes C_0}$, $V_0 = [T \cdot H \cdot W]$ and $A_0[H \cdot W \cdot t + W \cdot i + j] = \mathcal{F}[t, i, j, :]$ for $t \in [T], i \in [H], j \in [W]$, and call $\mathcal{F}$ the base of $\mathcal{G}$. Due to the definition of $\mathcal{G}$, the nodes $V_l$ at any layer define a partition of the base tensor, defined by associating to $v \in V_l$ the set $p(v) = \{(i, j)| \bigcirc_{l' < l} P_{l'}((i, j)) = v\}$. We call the map $S_l : v \mapsto p(v)$ the *spatial registration* at layer $l$. Thus, the set of SRs $\{S_l|l = 1, ..., L\}$ forms a hierarchical segmentation of the base tensor and the input scene from which it is constructed. An intuitive depiction of an example PSG is shown in Fig. 1. The base tensor itself is considered level-0 of a PSG, containing $|V_0| = T \cdot H \cdot W$ vertices, the trivial partition $S_0$ that assigns each spatiotemporal position $(t, h, w)$ to its corresponding vertex, and the attribute vectors defined by indexing into the base tensor, $A_0 : (t, h, w) \mapsto \mathcal{F}[t, h, w, :]$. By convention, we always prepend to the attribute vector $A_l(v)$ of a vertex $v$ its relative time index $t^v \in [T]$ and the image centroid $(c_h^v, c_w^v)$ of its corresponding segment in $S_l$.

## S2 PSGNet Architecture

The PSGNet architecture consists of three stages: feature extraction, graph construction, and graph rendering. In the first stage, a spatially-uniform feature map is created for any input movie by passing it through a Convolutional Recurrent Neural Network (ConvRNN) $\mathcal{F}_{\Theta_0}$. The tensor of feature activations from one convolutional layer of the ConvRNN is then used as the base tensor for constructing a spatiotemporal PSG. Taking these features as the "Level-0" nodes, the higher levels of a PSG are built one at a time by applying a learned Graph Constructor $\mathcal{GC}_{\Theta_1}$. This stage itself contains two types of of module, *Graph Pooling* and *Graph Vectorization.* Thus the final PSG representation of an input movie has $L + 1$ levels after applying $L$ (Pooling, Vectorization) pairs. Finally, the PSG is the passed through a decoder $\mathcal{R}$, which renders graph node attributes (and spatial registrations for the top level of graph nodes) into RGB, depth, normal, segment, and RGB-change maps for each frame of the input movie.

Formally, then, we define the parameterized class of neural networks $\mathcal{F}_{\Theta}^{\mathbf{PSG}}$ as functions of the form

$$\mathcal{F}_{\Theta}^{\mathbf{PSG}} = \mathcal{R} \circ \mathcal{GC}_{\Theta_1} \circ \mathcal{F}_{\Theta_0}, \tag{1}$$

where $\mathcal{F}$ is the ConvRNN-based feature extractor, $\mathcal{GC}$ is the graph constructor, $\mathcal{R}$ is the graph rendering, and $\Theta = \Theta_0 \cup \Theta_1$ are the learnable parameters of the system.

Note that the decoder does not have any learnable parameters of its own: it takes as input only the spatial registrations produced by Graph Pooling and the node attributes produced by Graph Vectorization, using them to "paint-by-numbers" (Quadratic Texture Rendering, **QTR**) or "draw shapes" (Quadratic Shape Rendering, **QSR**) as described in the main text. This strong constraint on decoding is what allows PSGNet to learn *explicit* representations of scene properties. Without additional parameters to convert a latent code into a rendered image, the latent codes of a PSG (i.e. the node attribute vectors) are optimized to have the same encodings as their (self-)supervision signals – color, depth, surface normal vectors, etc.

Below we describe each component of the PSGNet architecture in detail and give further background for its motivation. We then give the specific hyperparameter choices used in this work.

### S2.1 ConvRNN Feature Extraction

A convolutional recurrent neural network (ConvRNN) is a CNN augmented with both local recurrence at each layer and long-range feedback connections from higher to lower layers. Such local recurrent and long-range feedback connections are ubiquitous in the primate visual system, where it is hypothesized that they play a role in object recognition and scene understanding [23, 49, 52]. Large-scale neural network implementations of ConvRNNs optimized for ImageNet categorization have been shown achieve performance gains on a per-parameter and per-unit basis, and make predictions of neural response dynamics in intermediate and higher visual cortical areas [37]. Moreover, models with both local and long-range recurrence are substantially more parameter-efficient than feedforward CNNs on a variety of perceptual grouping tasks, which suggested that multiple forms of recurrence could provide useful inductive biases for scene decomposition [24, 34, 35, 14].

ConvRNNs prove useful in our application because they naturally integrate high- and low-level visual information, both of which are critical to understand scene structure. The bottom-up-and-top-down ConvRNN architecture is related to the standard U-Net structure (e.g. [4].) Long-range feedback in a ConvRNN plays a role analogous to the upsampling layers of a U-Net, while locally recurrent cells are similar to "skip-connections" in that they combine features of the same spatial resolution. However, the ConvRNN provides more flexibility than a U-Net in terms of how information from multiple layers is integrated. First, it can be unrolled for any number of "passes" through the full network (the operation of each feedforward, locally recurrent, and feedback connection), with more passes producing more nonlinear feature activations at each layer. Second, we can take advantage of this change in features over time by performing different stages of PSG construction after differing numbers of passes. In particular, the initial construction of Level-1 spatial registration from ConvRNN features (Level-0 nodes) yields the sharpest partition of the scene into "superpixel"-like segments when done after a single ConvRNN pass; this is because further perturbations to the ConvRNN features by recurrent local and long-range convolution tend to smooth them out, degrading the precise boundaries between scene elements. On the other hand, features obtained after multiple passes have integrated more mid- and high-level information about the scene – including across the larger spatial receptive fields found in higher ConvRNN layers – making them more accurate at predicting mid- and high-level PSG attributes, like depth and surface normals (data not shown.)

**Implementation.** Formally, let $F_k^p$ be the ConvRNN feature activations at layer $k$ of the backbone CNN after pass number $p$. Long-range feedback connections combine features from a higher layer $k + k'$ with this lower, potentially spatially larger layer $k$. This takes the form

$$\tilde{F}_k^p = ReLU(U_k^{k'} * Resize(F_{k+k'}^{p-1})), \tag{2}$$

where $U_k^{k'}$ is a convolution kernel and $Resize$ is bilinear upsampling to the resolution of the target layer. On the initial pass ($p = 0$), $F_{k+k'}^{p-1}$ are defined to be a tensor of zeros.

Local recurrent connections combine features within a CNN layer, which necessarily have the same spatial dimension:

$$F_k^p = Combine_{\{W_k, U_k\}}(F_{k-1}^p, \ F_k^{p-1}, \ \tilde{F}_k^p), \tag{3}$$

where $W_k$ and $U_k$ are, respectively, the feedforward and recurrent convolution kernels at that layer, $\tilde{F}_k^t$ are any features produced by feedback to that layer, and $Combine(a, b, c)$ is a nonlinear function that combines the convolved features, such as $ReLU(a + b + c)$. The functional form of $Combine$ defines a "Local Recurrence Cell" (such as a modified Vanilla RNN or LSTM Cell [37].) As with feedback, locally recurrent features $F^{p-1}$ are defined to be zero on the initial pass.

Thus, a ConvRNN architecture is fully specified by its backbone CNN, its list of feedback connections $[(k_0, k_0 + k_0'), (k_1, k_1 + k_1'), \ldots]$, the structure of its $Combine$ functions, and hyperparameters of all convolution kernels.

**In this work.** Here we use a backbone CNN with 5 feedforward layers. The layers have $(40, 64, 96, 128, 192)$ output channels, respectively, each followed by $2 \times 2$ max pooling with stride 2 to downsample the spatial dimensions by a factor of two per layer. There are feedback connections from all higher layers to the first layer, whose feature activations become the Level-0 PSG nodes. The locally recurrent "Cell" installed at each layer is what we call an EfficientGatedUnit (EGU), as it was inspired by the parameter-efficient, *feedforward* architecture EfficientNet [48] and designed to satisfy two properties: (i) learned gating of convolutional inputs, and (ii) pass-through stability, i.e. on the initial pass through the ConvRNN, outputs from the convolution operation are unaffected by the recurrent cell before being passed to the next layer. Together, these properties mean that local recurrence produces a dynamical perturbation to the feedforward activations, which was previously found to help performance on ImageNet classification [37]. The EGU $Combine$ function has the form

$$\overline{F}_k^p = F_k^{p-1} + \tilde{F}_k^p + ReLU(U_k * (W_k^{in} * F_{k-1}^p + F_k^{p-1})), \tag{4}$$

$$F_k^p = W_k^{out} * [\sigma(W_k^e * ReLU(W_k^r * <\overline{F}_k^p >)) \odot \overline{F}_k^p], \tag{5}$$

where equation (5) is a "Squeeze-and-Excitation" gating layer [20]: it takes mean of input features across the spatial dimensions, $< \overline{F} >$, reduces and expands their channel dimensions with the $1 \times 1$ kernels $W_k^r$ and $W_k^e$, then multiplicatively gates the input $\overline{F}$ with the sigmoid function $\sigma$ of this spatially broadcast tensor. The final layer output is a feedforward convolution of this tensor with the

kernel $W^o ut_k$. Other than $W_k^r$ and $W_k^e$, all feedforward kernels $W_k$ are $3 \times 3$; All local recurrent kernels $U_k$ are $5 \times 5$; and all feedback kernels $U_k^{k'}$ are $1 \times 1$. The "efficiency" of this cell comes from making the kernels $W_k^{in}$ and $U_k$ *depth-separable*, i.e. a convolution that operates per feature channel with the full spatial kernel dimensions followed by a $1 \times 1$ convolution that mixes channels. Following the original EfficientNet structure, we have the $W^{in}$ increase the channel dimension by a factor of 6 at each layer and the $W^r$ reduce it by a factor of 4. This fully determines the required output channel dimensions of all other convolution kernels.

Given a RGB movie $x$ written in channel-minor form with shape $(T, H_0, W_0, 3)$, we form the backward temporal difference $\delta x = x[1:T] - x[0:T-1]$ (where a block of 0s is padded on the end of $x[1:T]$ and the beginning of $x[0:T-1]$), and then form the channel-wise concatenation $\Delta(x) = x \oplus \delta x$, so that the input data layer has 3*2 = 6 channels. The ConvRNN is then run independently on each input frame of $\Delta(x)$ for $n_{\text{unroll}} = 3$ passes. We take outputs from the first convolutional layer in $\mathcal{F}_{\Theta_0}(\Delta(x))$ after each up-down pass, forming a tensor of shape $(T, n_{\text{unroll}}, H, W, C)$, where $H, W, C$ are the output dimensionalities of the Conv1 layer. This is used as input to the graph constructor. PSGNetS is trained on static images (i.e. movies of length $T = 1$) while PSGNetM is trained on movies of length $T = 4$; either model can be evaluated on movies of arbitrary length. All ConvRNN parameters are optimized end-to-end with respect to the **QSR** and **QTR** losses on the final PSG.

## S2.2 Learned Perceptual Grouping and Graph Pooling

The features extracted from ConvRNN layer $k$ after a chosen number of passes $p$ are considered, for each input movie frame $t$, the Level-0 vertex/node sets $V_0^t$. Thus $|V_0^t| = H \cdot W$ and the attribute map $A_0 : V_0 \to \mathbb{R}^C$ is defined as

$$A_0(v) = A_0(W \cdot i + j) = [t] \oplus [i, j] \oplus F_k[t, p, i, j, :], \tag{6}$$

that is, simply indexing into each spatial position of the base feature activation tensor and prepending the movie time step and spatial coordinates. At this stage, there are no within-layer edges or parent edges (because there are no higher-level nodes) and the spatial registration $R_0$ is the trivial partition of singleton nodes. As described in the main text, building a new set of child-to-parent edges $P_0$ from $V_0$ – or more generally, $P_l$ from $V_l$ – requires three computations: (1) a function $D_\phi$ (which may have learnable parameters) that assigns a nonnegative affinity strength to each pair of nodes $(v, w)$; (2) a thresholding function $\epsilon$ (which has no learnable parameters) that converts real-valued affinities to binary within-layer edges, $E_l$; and (3) an algorithm for clustering the graph $(V_l, E_l)$ into a variable number of clusters $|V_{l+1}|$, which immediately defines the child-to-parent edge set $P_l$ as the map that assigns each $v \in V_l$ its cluster index. These operations are illustrated on the left side of Figure S1.

Formally, we define a $k$-dimensional *affinity function* to be any symmetric function $D_\phi : \mathbb{R}^k \times \mathbb{R}^k \to \mathbb{R}^{\geq 0}$, parameterized by $\phi$, and a *threshold function* to be any symmetric function $\epsilon : 2^{\mathbb{R}} \times 2^{\mathbb{R}} \to \mathbb{R}^{\geq 0}$. Given a set graph vertices $V_l$, corresponding attributes $A_l : V_l \to \mathbb{R}^{k_l}$ we then construct the within-layer edges $E_l = \{\{v, w\} | D_\phi(A_l(v), A_l(w)) > \epsilon(D(v), D(w))\}$, where $D(v) = \{D_\phi(A(v), A(w')) | w' \in V_l\}$.

We use four pairs of affinity and threshold functions in this work, each meant to implement a human vision-inspired principle of perceptual grouping. These principles are ordered by increasing "strength," as they are able to group regions of the base tensor (and thus the input images) that are increasingly distinct in their visual features; this is because the higher principles rely on increasingly strong *physical assumptions* about the input. The stronger principles P3 and P4 are inspired by the observation that infants initially group visual elements into objects based *almost exclusively* on shared motion and surface cohesion, and only later use other features, such as color and "good continuation" – perhaps after learning which visual cues best predict motion-in-concert [45]. For graph clustering, we use the Label Propagation algorithm because it does not make assumptions about the number of "true" clusters in its input and is fast enough to operate online during training of a PSGNet. However, any algorithm that meets these requirements could take its place.

**Implementation.** The four types of affinity and thresholding function are defined as follows:

*Principle P1: Feature Similarity.* Let $\mathbf{v} := A_l(v)$ denote the attribute vector associated with node $v$ at graph level $l$ (excluding its time-indexing and spatial centroid components.) Then the P1 affinity between two nodes $v, w$ is the reciprocal of their L2 distance, gated by a binary spatial window

$$D^1(\mathbf{v}, \mathbf{w}) = \frac{\mathbb{1}(||c(v) - c(w)||_m < \delta_{\text{dist}})}{||\mathbf{v} - \mathbf{w}||_2}, \tag{7}$$

where $c(v)$ is the centroid of node $v$ in $(i, j)$ coordinates given by its spatial registration and $|| \cdot ||_m$ denotes Manhattan distance in the feature grid. The P1 affinities are thresholded by the reciprocal of their local averages,

$$\epsilon^1(D^1(v), D^1(w)) = \min\left(\frac{1}{\overline{D}^1(v)}, \frac{1}{\overline{D}^1(w)}\right), \tag{8}$$

where $D^1(v)$ denotes the set of nonzero affinities with node $v$ (i.e. affinities to nodes within its spatial window) and $\overline{D}^1(v)$ its mean. Thus P1 produces binary edges $D^1(\mathbf{v}, \mathbf{w}) > \epsilon^1$ between nodes whose attribute L2 distance is less than the spatially local average, without learned parameters.

*Principle P2: Statistical Co-occurrence Similarity.* This principle encodes the idea that if two nodes appear often in the same pairwise arrangement, it may be because they are part of an object that moves (or exists) as a coherent whole. Thus it is a way of trying to infer motion-in-concert (the ground truth definition of an object that we use in this work) without actually observing motion, as when visual inputs are single frames. This is implemented by making $D^2_{\phi_2}(\mathbf{v}, \mathbf{w})$ inversely proportional to the reconstruction error of a Variational Autoencoder (VAE, [26]) $H_{\phi_2}$, so that common node attribute pairs will tend to be reconstructed better (and have higher affinity) than rare pairs. Formally,

$$\mathbf{e}_{vw} := |\mathbf{v} - \mathbf{w}|, \tag{9}$$

$$\hat{\mathbf{e}}_{vw} := H_{\phi_2}(\mathbf{e}_{vw}), \tag{10}$$

$$D^2(\mathbf{v}, \mathbf{w}) = \frac{1}{1 + \nu_2 \cdot ||\mathbf{e}_{vw} - \hat{\mathbf{e}}_{vw}||_2}, \tag{11}$$

where $\nu_2$ is a hyperparameter and both $\mathbf{e}_{vw}$ and $\hat{\mathbf{e}}_{vw}$ are vectors of the same dimension as $\mathbf{v}$ and $\mathbf{w}$. This and all other VAEs in this work (see below) are trained with the beta-VAE loss [18],

$$\mathcal{L}_{\text{VAE}} = ||\mathbf{e}_{vw} - \hat{\mathbf{e}}_{vw}||_2 + \beta\mathcal{L}_{\text{KL}}(\hat{\mu}_{vw}, \hat{\sigma}_{vw}), \tag{12}$$

where $\beta$ is a scale factor and $\mathcal{L}_{\text{KL}}$ is the KL-divergence between the standard multivariate unit normal distribution and the normal distributions defined by the input-inferred $\mu$ and $\sigma$ vectors (VAE latent states.) The $D^2$ affinities are thresholded at $0.5$ for all node pairs to produce binary edges.

*Principle P3: Motion-driven Similarity.* Parts of a scene that are *observed* moving together should be grouped together, according to our physical definition of an object. This could be implemented by tracking nodes across frames (e.g. finding nearest neighbors in terms of nodes' attributes) and then defining co-moving nodes as those whose relative Eucldean distance (according to their $(x, y, z)$ attributes) changes less than some threshold amount. Because spatial and motion inference may have different amounts of observation noise for different nodes, though, we instead extend the VAE concept above to compute "soft" node affinities across and within frames. Let $H^w_{\phi^w_3}$ and $H^a_{\phi^a_3}$ be two new VAE functions and $\nu^w_3$ and $\nu^a_3$ two new hyperparameters. Then, just as above, these define affinity functions $D^{3w}(\mathbf{v}, \mathbf{w})$ and $D^{3a}(\mathbf{v}, \mathbf{u})$ between nodes within each frame ("spatial") and across adjacent frames ("temporal," i.e. $v \in V^t_l, u \in V^{t+1}_l$), respectively. The only difference between the P2 and P3 affinities (other than having separate parameters) is that the latter are trained and evaluated *only on nodes whose motion attributes*, $\Delta(v)$, *are greater than a fixed threshold.* This means that the P3 VAEs only learn which nodes commonly *move* together, not merely *appear* together. Grouping *via* the P3 edges (affinities thresholded at 0.5) then proceeds by clustering the entire spatiotemporal graph, $G_l = (\bigcup_t V^t_l, \bigcup_t(E^{3w}_t \cup E^{3a}_t))$.

*Principle P4: Self-supervised Similarity from Motion.* The final principle is based on the assumption that if two nodes are seen moving together, then nodes with similar attributes and relationships should be grouped together in future trials – even if those nodes are not moving. This implies a form of self-supervised learning, where the signal of motion-in-concert acts as an instructor for a separate affinity function that does not require motion for evaluation. In principle, any supervision signal could be used – including "ground truth" physical connections, if they were known – but here we assume that motion-in-concert provides the most accurate signal of whether two visual elements are linked. We therefore implement the P4 affinity function as a multilayer perceptron (MLP) on differences between node attribute vectors,

$$D^4(\mathbf{v}, \mathbf{w}) = \sigma(Y_{\phi_4}(|\mathbf{v}' - \mathbf{w}'|)), \tag{13}$$

Figure S1: Schematics of the generic process for building a new PSG level and of the actual PSGNet architectures used in this work. (**Left**) To build nodes $V_{l+1}$, existing nodes $V_l$ (here shown as their spatially registered color rendering) are passed through pairwise affinity and thresholding functions to *Add Edges* $E_l$. The resulting graph $(V_l, E_l)$ is clustered according to *Label Propagation*; here nodes are colored by their assigned label (cluster index) at each iteration, which is updated by taking the most common label among graph neighbors. (Ties are resolved randomly.) After $M$ iterations, the kernels for *Pooling* nodes are defined by the final cluster assignments. This determines the child-to-parent edges $P_l$ and the new spatial registration $R_{l+1}$. Graph Vectorization, which constructs the new attributes $A_{l+1}$, is not shown here. (**Right**) The set of operations and perceptual grouping principles used to build PSGs in PSGNetM and PSGNetS (red outline.) Other than grouping by motion-in-concert (P3), all grouping constructs per-frame PSG nodes $V_l^t$. P3 builds Level 2M nodes that extend temporally across all input frames. *Extr* indicates ConvRNN feature extraction.

where $\mathbf{v}'$ is the attribute vector of node $v$ with its motion attribute $\Delta(v)$ removed, $Y_{\phi_4}$ is an MLP, and $\sigma(\cdot)$ is the standard sigmoid function, which compresses the output affinities into the range $(0, 1)$. The MLP weights are trained by the cross-entropy loss on the self-supervision signal from P3,

$$\mathcal{L}_{P4} = \sum_{v,w} \mathbf{CE}(D^4(\mathbf{v}, \mathbf{w}), \mathbb{1}(P_3(v) = P_3(w))), \tag{14}$$

where the "ground truth" is the indicator function on whether $v$ and $w$ have the same parent node, $P_3(v) = P_3(w)$, according to a round of P3 grouping from motion. The binary P4 edges are obtained by thresholding the affinities at 0.5.

*Label Propagation.* To construct new parent edges, nodes are clustered according to within-layer edges from one of the four affinity functions using the standard Label Propagation (LP) algorithm [47] (Fig. S1 left, middle column.) This algorithm takes as input only the edges $E_l$, the number of nodes at the current graph level $|V_l|$, and a parameter setting the number of iterations. Each node is initialized to belong in its own cluster, giving a set of labels $[|V_l|]$. Then for $q > 0$, iteration-$q$ labels are produced from iteration $q - 1$-labels by assigning to each node $v \in V_l$ the most common stage-$q - 1$ label among nodes connected to $v$ by an edge. Ties are resolved randomly. Let $P_l : V_l \to [m]$ denote the final converged label-prop assignment of nodes in $V$ to cluster identities, where $m$ is the number of clusters discovered. Effectively, the new child-to-parent edges $P_l$ define *input-dependent pooling kernels* for $V_l$ (Fig. S1 Left, right column.) The final stage of building a new PSG level, *Graph Vectorization*, uses these pooling kernels to aggregate statistics over the resulting partition of $V_l$, resulting in new nodes $V_{l+1}$ with $|V_{l+1}| = m$ (see below.)

**In this work.** Two PSGNet architectures are used in this work: PSGNetS, which sees only static images (single-frame movies), and PSGNetM, which sees true movies and therefore can detect and learn from object motion. PSGNetS builds a three-level PSG by applying P1 grouping on the Level-0 nodes (ConvRNN features) to build Level-1 nodes, followed by P2 grouping on the Level-1 nodes to build Level-2 nodes (Fig. S1 right, red outline.) PSGNetM builds a *branched* four-level PSG by adding additional rounds of grouping to PSGNetS: P3 grouping from motion-in-concert builds a set of *spatiotemporal* Level-2 nodes (called Level-2M) from the Level-1 nodes; and these self-supervise a round of P4 grouping on the original Level-2 nodes to build Level-3 nodes (Fig. S1 right.) Other PSGNet architectures could build PSGs with different hierarchical structures by applying these (or

new) grouping principles in a different pattern. For instance, models that grouped from motion at different scales could, in principle, learn hierarchical decompositions of objects into their flexibly moving parts [56].

The VAEs used in P2 and P3 grouping are MLPs with hidden layer dimensions $(50, 10, 50)$; the 10 dimensions in the latent state are considered as 5 pairs of $(\mu, \sigma)$ parameters for independent normal distributions. The $\beta$ scaling factor for each KL loss is 10. The slope hyperparameters were set at $\nu_2 = 3.5$, $\nu_3^w = \nu_3^a = 10.0$ by grid search on a validation subset of **Playroom**. The P4 MLP used in PSGNetM has hidden layer dimensions $(250, 250)$ and an output dimension of 1. In the P2 binary affinity function variant (BAff, Table 2 [21]), affinities are predicted by an MLP with hidden layer dimensions $(100, 100)$ and an output dimension of 1. All applications of LP proceed for 10 iterations, which is enough to reach convergence (i.e., unchanging label identity with further iterations) for the majority of nodes.

## S2.3 Graph Vectorization

Given nodes $V_l$ and pooling kernels defined by $P_l$ from the previous stage, Graph Vectorization constructs a new level of nodes $V_{l+1}$ and their associated attributes $A_{l+1} : V_{l+1} \to \mathbb{R}^{C_{l+1}}$. This also specifies a new spatial registration $R_{l+1}$, which maps each spatial index into the Level-0 nodes, $(i, j)$, to the index of its unique parent in $V_{l+1}$. We call this process "Vectorization" because it encodes the group properties of a set of entities (here, lower-level graph nodes) as components of a single attribute vector.

New node attributes are created in two ways: first, through permutation-invariant aggregation functions of the lower-level nodes in each segment, and second, by applying MLPs to these aggregates. The reason for computing the latter set of attributes is that aggregate statistics are unlikely to provide the direct encoding of scene properties (color, depth, normals, etc.) needed for Graph Rendering. The use of MLPs allows these direct encodings to be learned as nonlinear functions of the aggregates, with parameters shared across nodes (much as standard convolution kernels share parameters across spatial positions.)

**Implementation.** For a vertex $v \in V_{l+1}$, let $P_l^{-1}(v)$ denote the set of vertices in $V_l$ whose parent is $v$, according to the Graph Pooling operation; let $Seg_{l+1}[v]$ denote the set of spatiotemporal positions in the base tensor $\mathcal{F}$ that belong to the segment associated with $v$, that is $\{(t, h, w) \,|\, S_{l+1}[t, h, w] == v\}$. Then the Graph Vectorization module computes the new attribute vector $A_{l+1}^{\mathbf{agg}}(v)$ as the summary statistics of $A_l$ over the domain $P_l^{-1}(v)$, concatenated with the summary statistics of $A_0 \equiv \mathcal{F}$ over the domain $Seg_{l+1}[v]$. This process is depicted in Fig. 1. In the context of standard neural network architectures, aggregation can be understood as various pooling operations over the *input-dependent pooling kernels* induced by $P_l$ and $S_{l+1}$, rather than over typical fixed-size kernels (e.g. $2 \times 2$ windows on the feature grid.) Though the size of the input kernels is variable, the output attribute vectors have a fixed number of components because each summary statistic maps a set of attributes in $A_l$ to a single number.

For aggregation, we compute the mean of the first and second power of each lower-level node attribute as described in the main text. In addition to aggregating over the set $Seg_{l+1}[v]$ for each node, we also compute aggregates over nine subsets of $Seg_{l+1}[v]$ defined with respect to spatial slices of the registration $S_{l+1}$: its boundary, i.e. the set of nodes $w \in Seg_{l+1}[v]$ whose registrations in the base tensor are adjacent to those of a node from a different segment $Seg_{l+1}[v'], v' \neq v$; its four "quadrants," the subsets of nodes whose registered centroids are above and to the right, above and to the left, etc., of the centroid of $Seg_{l+1}[v]$; and the four boundaries of those quadrants. Thus if nodes in $V_l$ have $C_l$ attribute components (including their spatial centroids), the new aggregates $A_{l+1}^{\mathbf{agg}}(v)$ will have $2 \cdot 10 \cdot C_l$.

Two MLPs produce new attributes from these aggregates: A "unary" function $H_{l+1}^U(A_{l+1}^{\mathbf{agg}}(v))$, which operates on each aggregate in $V_{l+1}$; and a "binary" function $H_{l+1}^B(|A_{l+1}^{\mathbf{agg}}(v) - A_{l+1}^{\mathbf{agg}}(w)|)$, which operates on pairs of aggregates. (This latter operation is a type of graph convolution on the fully-connected graph of the Level-$l + 1$ aggregates. Both operations can be merged into one by doing graph convolution with self-edges, but the implementation is simpler when keeping them separate.) The final set of new attributes $H^{\mathbf{new}}(v)$ is given by

$$H_{l+1}^{\mathbf{new}}(v) = H_{l+1}^U(v) + \frac{1}{|V_{l+1}|} \sum_{w \in V_{l+1}} H_{l+1}^B(v, w), \tag{15}$$

that is, by summing the unary MLP outputs and the mean of the binary MLP outputs across node pairs. The aggregate attributes and the learned attributes are concatenated to give the full attribute labeling maps, $A_{l+1} : v \mapsto A_{l+1}^{\mathbf{agg}}(v) \oplus H_{l+1}^{\mathbf{new}}(v)$.

**In this work.** All MLPs used for Graph Vectorization have hidden layers with dimensions $(100, 100)$. The number of output dimensions is based on how many attributes are needed for rendering (**QSR** and **QTR**). Each channel of an output **QTR** feature map requires 6 components, and each of 6 **QSR** constraints requires 4 components. Because the full vectorization process would cause the number of node attributes to grow geometrically in the number of PSG levels, we compute only mean attributes and perform only constant texture rendering (requiring a single component per output channel) for Level-1 nodes; in practice, these nodes have "superpixel"-like spatial registrations with little feature variation across their aggregation domains and with similar shapes.

## S2.4   Decoding

Two types of decoder render PSG nodes into images, allowing for (self-)supervision on other image-like tensors. These decoders both have fixed functional forms, where the rendered outputs depend only on the input PSG components. Since the decoders do not have learnable parameters, no information would be lost in performing downstream tasks with the PSG encodings themselves, rather than the decoder outputs. The lack of parameters also forces all learning to happen in the PSG encoder and forces the PSG nodes to represent scene properties with a direct, explicit encoding (rather than an entangled, nonlinear function of an abstract latent state.)

*Quadratic Texture Rendering.* Given the nodes and attributes of layer of a PSG, $V_l, A_l$, together with the layer's spatial registration $S_l$, quadratic texture rendering (**QTR**) creates an image by inpainting the value of an attribute for node $v$ onto the pixels in $Seg_l[v]$. However, rather than paint uniformly throughout $Seg_l[v]$, QTR paints a quadratic function of attributes. Let $a, a_h, a_w, a_{hh}, a_{ww}$, and $a_{hw}$ denote six attribute dimensions from $A_l(v)$, and let $(c_h^v, c_w^v)$ denote the centroid of $Seg_l[v]$ in image coordinates. Then define the quadratic form $qtr[v](i,j) = a + a_h(i - c_h^v) + a_w(j - c_w^v) + \frac{1}{2}a_{hh}(i - c_h^v)^2 + \frac{1}{2}a_{ww}(j - c_w^v)^2 + \frac{1}{2}a_{hw}(i - c_h^v)(j - c_w^v)$. The (single-channel) image rendered from these attributes is then given by $\mathbf{QTR}_l^a : (i,j) \mapsto qtr[S_l[i,j]](i,j)$. Rendering an RGB image with a **QTR** decoder therefore takes 18 components of $A_l$. See Fig. 1 and S3 for examples of **QTR**.

When rendering a **QTR** that will be supervised with a depth map, we distinguish two additional attributes $x$ and $y$ at each graph level so that its nodes will have a representation as a 3D point cloud. To compute a projective 2D-3D self-consistency, we treat our system as a pinhole camera. We compute the loss $\mathcal{L}_{\mathbf{proj}}(X) = \sum_{v \in V_l(X),t} ||\mathbf{proj}[a_x^t(v), a_y^t(v), a_z^t(v)] - (c_h^{v,t}, c_w^{v,t})||_2$ where $a_z^t(v)$ is the predicted depth (from the depth-supervised channel $z$) and $\mathbf{proj} : \mathbb{R}^3 \to \mathbb{R}^2$ denotes the pinhole camera perspective projection. The camera focal length parameter is provided on a per-dataset basis in training.

*Quadratic Shape Rendering.* A Quadratic Shape Rendering (**QSR**) decoder renders predictions of what 2D silhouette a PSG node produces in the input scene, elaborating on a procedure developed in [7] to "draw" a shape as the intersection of signed distance function constraints. Let $D$ be the number of constraints and $p_x^d, p_y^d, p_\rho^d, p_\alpha^d, d \in [D]$ be $4D$ components from $A_l(v)$. For each $d \in [D]$, let $qsr^d[v](i,j)$ be the scalar field defined by taking normalized signed distance of point $(i,j)$ to the locus of a 2D parabola, i.e. $qsr^d[v](x,y) = \sigma\left(p_\alpha^d[y\cos(p_\rho^d) - x\sin(p_\rho^d) - p_x^d]^2 - [x\cos(p_\rho^d) + y\sin(p_\rho^d) - p_y^d]\right)$ where $\sigma$ is the standard sigmoid function. Let $qsr[v](i,j) = \min_{d \in [D]} qsr^d[v](i,j)$. Define the segment map $\mathbf{QSR}_l : (i,j) \mapsto \arg\max_{v \in V_l} qsr[v](i,j)$ and the segment "confidence map" $\mathbf{QSR}_l^c : (i,j) \mapsto \max_{v \in V_l} qsr[v](i,j)$. In this work, we use $D = 6$. See Fig. 1 and S3 for examples of **QSR** and note that the rendered shapes do not depend directly on the segmentations $S_l$, unlike in **QTR**. This means that the attributes vectors of a PSG are sufficient to generate an image *via* **QSR**, a property we use below to demonstrate the symbolic structure of the PSG representation.

*Training Losses.* For the reconstruction of RGB and prediction of depth and normal maps, we distinguish at each layer six **QTR** attribute dimensions for each attribute $\mathbf{a} \in \{R, G, B, z, N_x, N_y, N_z\}$. The associated loss is $\mathcal{L}_{\mathbf{a}}(\theta, X) = \sum_{i,j,l,t}(\mathbf{QTR}_{\theta,l,t}^{\mathbf{a}(X)}(i,j) - \mathbf{gt}_c(X)(i,j))^2$, where $\mathbf{gt}_{\mathbf{a}}(X)$ is the ground-truth value of attribute $\mathbf{a}$ associated with input $X$. For reconstructing RGB difference magnitude when movies are given, we distinguish a channel $\Delta$ (and its associated QTR attributes), self-supervising on $|\sum_{r,g,b}(\delta X)^2|$ for ground-truth. We also compute losses for QSR predictions,

taking the softmax cross-entropy between the shape map for each predicted object and the self-supervising "ground truth" indicator function of whether a pixel is in that object's segment, i.e. $\mathcal{L}_{\mathbf{QSR}}(\theta, X) = \sum_{t,i,j,v \in V_L(X_t)} \mathbf{SoftMaxCE}(qsr[v](i,j), \mathbb{1}[(i,j) \in S_L(v)])$.

## S3 Datasets, Baseline Models, and Evaluation Metrics

### S3.1 Datasets

**Primitives.** This dataset was generated in a Unity 2019 environment by randomly sampling 1-4 objects from a set of 13 primitive shapes (Cube, Sphere, Pyramid, Cylinder, Torus, Bowl, Dumbbell, Pentagonal Prism, Pipe, Cone, Octahedron, Ring, Triangular Prism) and placing them in a square room, applying a random force on one object towards another, and recording the objects' interaction for 64 frame-long trials. The textures and colors of the background, lighting, and each object were randomly varied from trial to trial. The training set has in total 180,000 frames with 2 and 3 objects. The test set has in total 4,000 frames with 1-4 objects. Each frame includes an image, a depth map, a surface normals map, a segmentation map of the foreground objects (used only for evaluation), and the camera intrinsic parameters (which are constant across each dataset.) During training and evaluation of MONet, IODINE, and PSGNetS, individual frames across different trials are randomly shuffled, so object motion is not apparent from one frame to the next.

**Playroom** This dataset was generated using the same Unity 2019 environment as **Primitives** with two differences: first, the floor and walls of the room were given naturalistic textures (including natural lighting through windows in the walls and ceiling); and second, the objects were drawn from a much larger database of realistic models, which have naturalistic and complex textures rather than monochromatic ones. For each of 1000 training trials or 100 test trials, 1-3 objects were randomly drawn from the database and pushed to collide with one another. Each trial lasts 200 frames, giving a total of 200,000 frames in the training set and 20,000 in the testing set. 100 of the training trials were randomly removed to be used as a validation set. When presented to PSGNetM and OP3, multi-frame clips from each trial were randomly selected and shuffled with clips from other trials, so object motion was apparent in a subset of training examples. Both datasets will be made available upon request, and the Unity 2019 environment for generating these data (ThreeDWorld) will be available upon publication.

**Gibson.** We used a subset of the Gibson 1.0 environment[*]. The subset is scanned inside buildings on the Stanford campus, and is subdivided into 6 areas. We used area 1, 2, 3, 4, 6 for training, half of area 5 for validation, and the other half of area 5 for testing. The training set has 50,903 images and the validation and test set each have 8,976 images, along with depth maps, surface normals maps (computed approximately from the depth maps), full-field instance segmentation labels, and the camera intrinsic parameters.

**CLEVR6 and MultiDSprites.** For zero-shot transfer experiments, we tested PSGNets and baselines on two datasets used in the unsupervised object discovery literature: CLEVR and MultiDSprites (MDS) [4, 17]. We evaluated each **Playroom** or **Primitives**-pretrained model on the first 2000 examples of each dataset, in the case of CLEVR evaluating only on scenes with six ("CLEVR6") or fewer objects as in [17]. Qualitatively, the CLEVR dataset is similar to our **Primitives** but with no background scene components and with a narrower range of object shapes; MDS consists of 2D, monochromatic shapes that sometimes overlap each other on a solid background.

### S3.2 Model and Training Implementations

We implemented MONet, IODINE, and OP3 as described in the original papers and publicly available code [4, 17, 54]. We used the same convolutional encoder and decoder for all three models. The encoder has 4 layers with (32, 32, 64, 64) channels each. The decoder also has 4 layers with (32, 32, 32, 32) channels each. The number of object slots $K$ is set to 7 for all models on the synthetic datasets and 12 for **Gibson**. MONet uses a 5-block U-Net for attention. It has in total 14.9M parameters. IODINE and OP3 use 5-step iterative inference [17, 54]. IODINE has 1.7M parameters. OP3 has 1.6M parameters. PSGNetS has 1M parameters and PSGNetM 1.3M.

For Quickshift++, we used 1000 images from each training set to search for the best hyperparameters. Table S1 shows the hyperparameters we found for each dataset.

**Training.** We trained baseline models with the Adam optimizer [27] with learning rate 0.0001 and batch size 128. Gradients with norm greater than 5.0 were clipped. MONet was trained with 4

---

[*]http://buildingparser.stanford.edu/dataset.html

Table S1: Quickshift++ hyperparameters for each dataset.

| Dataset | $k$ | $\beta$ |
|---|---|---|
| Primitives | 20 | 0.9 |
| Playroom | 20 | 0.95 |
| Gibson | 80 | 0.95 |

Titan-X GPUs. IODINE and OP3 were each trained with 8 GPUs. The training took between 48 and 96 hours to converge, depending on the model and dataset. PSGNetS and PSGNetM each were trained with batch size 4 on a single Titan-X GPU for 24 hours using the Adam optimizer [27] with learning rate 0.0002. All models are based on CNN backbones, so can take images of any size as input; however, training on images $> 64 \times 64$ was computationally prohibitive for the baseline models, so we trained all models and report all evaluation metrics on images of this size. PSGNets take significantly less time and resources to train than the baselines, so for visualization we trained and evaluated versions of PSGNetS and PSGNetM on $128 \times 128$ images. This increases segmentation metric performance by 5-10% (data not shown.) To measure the effect of random weight initialization, we developed a PSGNetM-RGB model and trained five copies on movies from **Playroom** with different random seeds. These models scored (mean +/- stdev) of (**Recall** 0.65 +/ 0.01, **mIoU** 0.59 +/- 0.01, **BoundF** 0.60 +/- 0.01) on held-out images, suggesting that performance is quite consistent across random seeds. We therefore report only the performance of single trained models in the rest of this work. Tensorflow [1] code for training and evaluating PSGNet models is available at github.com/neuroailab/PSGNets. Because PSGs can have a variable number of nodes per input scene at any level above the base tensor, but Tensorflow code generally operates on rectangular arrays, we distinguish a single "valid" attribute vector component at each PSG level that indicates whether a given node is real (part of the PSG) or being masked out.

### S3.3 Evaluation

We use standard metrics for object detection and segmentation accuracy with minor modifications, due to the fact that all tested models output full-field scene decompositions rather than object proposals: each pixel in an input image is assigned to exactly one segment. The **mIoU** metric is the mean across ground truth foreground objects of the *intersection over union* (IoU) between a predicted and ground truth segment mask (a.k.a. the Jaccard Index.) Because one predicted segment might overlap with multiple ground truth segments or *vice versa*, for this and the other metrics we found the optimal one-to-one matching between predictions and ground truth through linear assignment, using $1.0 - $ **mIoU** as the matching cost. The **Recall** metric is the proportion of ground truth foreground objects in an image whose IoU with a predicted mask is $> 0.50$. **BoundF** is the standard F-measure (a.k.a. F1-score) on the ground truth and predicted boundary pixels of each segment, averaged across ground truth objects. **ARI** (Adjusted Rand Index) is the Permutation Model-adjusted proportion of pixel pairs correctly assigned to the same or different segments according to the ground truth segmentation, as used in [17] – except that we do not restrict evaluation to foreground objects. Linear assignment, ARI, and segment boundary extraction are implemented in Scikit-learn [40].

In the main text we report metrics on each test set for single model checkpoints, chosen by where **Recall** on each validation set peaks (within 400 training epochs for the baselines and 5 training epochs for PSGNets.) Variation in PSGNet performance due to random weight initialization was <2% across 5 models.

## S4 Comparing PSGNet to Baseline Models

Here we further describe the qualitative and quantitative differences between PSGNets and the CNN baselines as unsupervised scene decomposition methods. At a high level, these methods produce such different results because (1) they make different assumptions about how scenes are structured and (2) they use different architectures (and therefore have different "inductive biases") to learn segmentations. The CNN-based models MONet [4], IODINE [17], OP3 [54], and related methods [9] all assume that the visual appearance of a scene is generated by the combined appearances of up to $K$ latent factors. In essence, each model tries to infer the the parameters of the latent factors using a CNN encoder (or "de-renderer") and then reconstructs the scene with a decoder ("renderer") that operates on each factor and combines their results into an output image. These models therefore learn which scene components (i.e., segments of scenes with a certain 2D shape and appearance) are

Figure S2: Examples (one per row) of the CNN-based models' RGB, normals, and segmentation predictions on **Playroom**. (All models also receive supervision and make predictions in the depth channel, not shown here.) Unlike on **Primitives**, the baselines generally do not capture the shapes and textures of the objects in this dataset.

Figure S3: Six examples of PSGNetS top level (Level2) quadratic texture renderings of color (QTRs) and quadratic shape rendering (QSRs) on **Primitives**. Here the QSRs are filled in with the color of their associated nodes' color attributes, but they could be filled in with any other attribute. The QSRs are able to capture many of the simple silhouettes of objects in this dataset (compare to th QTRs, which are colored according to the Level2 unsupervised segmentations of each image.) Some silhouettes, such as that of the dumbbell (middle row, right column) are approximated by simpler shapes; adding additional quadratic constraints to the QSRs could produce more complex shapes.

common across a dataset; over the course of training, they get better at detecting and reconstructing these components.

In contrast, PSGNets do not learn *via* the assumption that scenes can be decomposed into the appearance of several latent factors. They instead assume that *pairs of visual elements are physically connected*, and therefore can be represented as parts of a single, higher-level entity. This difference in assumptions is subtle but yields a dramatically different optimization problem for PSGNets, because they do not need to learn which object silhouettes (segmentation masks) or global appearances are common across a dataset; the only need to learn which pairwise relationships appear frequently or persist through time (e.g. during object motion.) Whereas silhouettes and global appearances vary widely even for the same object across different views and contexts, pairwise relationships (such as differences in color, texture, or surface normal vectors) are highly constrained by the physics of real, solid objects: many are made of similar material and have smoothly changing shape across broad

Figure S4: Six examples of PSGNetM renderings and segmentations on **Playroom**. Only the top-level (Level3) quadratic texture renderings and spatial registration are shown, along with the ground truth for each input image. Note that the ground truth segmentations are not provided during training and only the color image is input to the model. The main failure mode of PSGNetM is undergrouping of large, static regions (e.g. first row, right column) or of objects with regions of very different textures (e.g. second and third rows, right column.)

regions of their surfaces. In cases where objects have visually and geometrically distinct subregions, their concerted motion should reveal the "ground truth" of their underlying cohesion.

In addition to this difference in assumptions for learning, PSGNets have a major architectural difference with CNN-based methods: they produce segments by *spatially non-uniform grouping of scene elements* rather than fixed-size convolution kernels whose weights are shared across visual space. It has previously been recognized that spatially uniform convolutions are poorly suited to produce sharp scene segmentations, motivating post-processing algorithms like conditional random fields for sharpening CNN outputs [5]. At heart, this is because region borders and interiors tend to have different CNN feature activations (as spatially uniform receptive fields inevitably cross boundaries) even when they have the same visual appearance. This problem is greater in the unsupervised setting, where there are no segmentation labels to indicate exactly where a boundary occurs. Thus, we hypothesized that the perceptual grouping mechanisms of PSGNets would produce sharper boundaries and be less prone to overfitting common object shapes than the CNN-based methods. This is consistent with the large previous body of work on *unlearned* graph-based perceptual grouping and scene segmentation (e.g. [44, 2]). We extend the ideas behind hierarchical graph-based segmentation by learning image features and affinity functions according to physical properties of scenes.

**Qualitative comparison of model error patterns.** Each model makes characteristic errors that can be explained by their respective architectures and loss functions. MONet reconstructs the input image (and here predicts depth and normals feature maps) by producing one segmentation mask at a time from a U-Net CNN [4], but imposes no further constraints on the structure of its outputs or latent states. As a result its segmentations and predictions can have high spatial resolution and low pixelwise reconstruction error even when inferring clearly non-physical "objects," such as segments split across disconnected regions of the image; moreover, it learns to lump together disparate background

Table S2: Training epochs before peak validation set object recall.

| Dataset | MONet | IODINE | OP3 | PSGNetS | PSGNetM |
|---|---|---|---|---|---|
| Primitives | 200 | 350 | - | 2.0 | - |
| Playroom | 38 | 83 | 24 | 1.8 | 1.0 |

regions because the expressive U-Net decoder can easily output single segment feature maps that adequately reconstruct predictable chunks of the scene (Fig. S2, second column.)

IODINE (and its dynamics-modeling successor OP3) instead jointly infers all the parameters of all scene latent factors in parallel and "spatially broadcasts" them over an intermediate feature map at the same resolution as its output [17]. This encourages each predicted "object" to occupy a single, spatially contiguous region, but also appears to degrade information about its shape; most foreground segment predictions resemble "blobs" that lack the distinguishing features of the shapes they represent (Fig. S2, third and fourth columns.)

Finally, PSGNetS segments the scene into regions that are both contiguous and high-resolution, but sometimes oversegments – that is, *undergroups* during learnable graph pooling. This is especially apparent for scene elements that are large (likely because learned affinities tend to fall off with distance) or contain strong shadows or reflections (because pairwise affinities may be low across a large change in appearance; see Figs. S3 and S4.) Some of these errors are reduced in PSGNetM, which can group visually distinct regions that have been observed moving together. However, the addition of P3 and P4 in this model can lead to overgrouping when two scene elements with similar appearance are or have been seen moving together (see Fig. 2C in the main text.) These two principles also do not provide learning signals about how to group scene elements that rarely or never move (such as floors and walls.) In future work, we will explore additional grouping principles that can apply to these scene elements and can better distinguish moving objects even when they share similar appearance.

**Learning efficiency and generalization.** Beyond the gap between PSGNets and CNN baselines in segmentation performance, the substantially greater learning efficiency and generalization ability of PSGNets suggest that their bottom-up perceptual grouping architecture is better suited for scene decomposition than the latent factor inference architectures of the baselines. On **Primitives**, where all models were able to decompose scenes reasonably well, peak object detection performance (**Recall**) on the validation set occurred after 200 and 350 training epochs for MONet and IODINE, respectively, but after only 2 epochs for PSGNetS (Table S2.) The difference was less pronounced when training on **Playroom**, where MONet, IODINE, and OP3 peaked after 38, 83, and 24 epochs – compared to 1.8 and 1.0 for PSGNetS and PSGNetM; however, the poor performance of the baselines on this dataset (<0.30 test set **Recall**) makes their learning efficiency a less important metric. For the same reason, it is hard to assess the across-dataset generalization of the baselines, since they all underfit one of the two synthetic datasets used in this work. Given that none of them achieved >0.10 **Recall** when tested on the dataset they were not trained on, we consider their generalization ability limited. In contrast, both **Primitives**-trained PSGNetS and **Playroom**-trained PSGNetM achieved >50% of their within-distribution test performance on their converse datasets, suggesting that a significant portion of their learned perceptual grouping applies generically to objects that have never been seen before. This is our expectation for algorithms that purport to learn "what objects are" without supervision, rather than learning to detect a specific subset of objects they have seen many times. Future work will explore what constitutes an optimal set of training data for the PSGNet models.

The CLEVR6 and MDS datasets, which were used to develop MONet and IODINE [4, 17], do not have the structured background or nearly the variety of object shapes as the TDW or Gibson datasets (though individual CLEVR objects are qualitatively similar to **Primitives** objects, with greater photorealism.) Thus, we expected that these datasets would be somewhat easier to segment than the ones we developed our models on. This was born out both by the PSGNets' zero-shot transfer performance (see main text) and by training on the datasets, where PSGNetS-RGB (with no further architecture modifications) reached recall of 0.73 on CLEVR6 and 0.80 on MDS – higher than on any of the TDW or Gibson datasets. Interestingly, the MONet and IODINE models did even better when trained directly on CLEVR and MDS, reaching recalls of (0.80, 0.84) on the former and (0.94, 0.96) on the latter. On the one hand, this could indicate that PSGNets would perform better on these much smaller and simpler datasets if their architectures were further constrained to avoid overfitting (as was required to find the NoDN* and PSGNet-RGB architectures.) On the other hand,

the fact that PSGNets perform almost equally well on these datasets *without ever seeing them* – and dramatically outperform baselines on more realistic images – demonstrates their *lack* of overfitting to dataset-specific objects, which we see as a relative strength of our approach. Nevertheless, there may be fruitful ways to combine the mask-prediction architectures of these baselines with the perceptual grouping inductive biases of PSGNets to get the best of both.

**Assessing the dependence of baselines on geometric feature maps.** Finally, because the CNN baselines were all developed to solve an autoencoding (or future prediction) problem on RGB input only, we wondered whether the task used in this work – reconstruction of supervising RGB, depth, and normals maps – might have have stressed these models in unintended ways (even though they were given strictly more information about input scenes during training.) Surprisingly, neither giving depth and normal maps as *inputs* (in addition to the RGB image) nor reconstructing RGB alone substantially changed the performance of MONet or IODINE on **Primitives**: both autoencoding variants of both models yielded performance within 10% of what they achieved on the original RGB to RGB, depth, and normals task. It is unclear why the baseline methods do not learn to decompose scenes better when given more information about their geometric structure (cf. the NoDN PSGNetM ablation.) One possible explanation is that color cues alone are highly predictive of object boundaries in **Primitives**, whereas the depth and normals channels only rarely indicate boundaries that are not easily detected in the color channels.

## S5 Visualizing PSGNet Outputs

Here we provide additional examples of components of predicted PSGs. Figure S3 compares the PSGNetS top-level (Level-2) color QTRs to the top-level QSRs on images from **Primitives**. The former are colored by quadratic "painting by numbers" in the Level-2 spatial registration inferred for each image, so better reconstruct both high-resolution details and smooth changes in color across large regions. The latter are shapes "drawn" by intersecting 6 predicted quadratic constraints per node, colored by each node's RGB attribute.

Figure S4 shows PSGNetM top-level (Level-3) QTRs for color, depth, and normals, as well as the top-level spatial registration, for single images in **Playroom**. Despite the wide variety of object sizes, shapes, and textures in this dataset, PSGNetM is largely able to segment and reconstruct the attributes of most. Undergrouping (oversegmentation) failures tend to occur when objects have sharp internal changes in color. Interestingly, human infants seem not to perceptually group visual elements by color in their first few months of life, relying instead on surface layout and motion [45]; it is possible that PSGNetM could better learn to group parts of these complex objects by explicitly ignoring the color and texture attributes of their nodes, which is feasible because of their direct and disentangled encoding.