[Reviews · NeurIPS 2020]

Review 1

Summary and Contributions: The paper proposes a model for learning of scene representations capable of producing object-centric representations without direct supervision from objects or segmentation maps by using supervision from the reconstruction of the RGB, depth and normals. The model starts by extracting visual features with a ConvRNN in order to have long-range connections. A graph representation is constructed in a hierarchical manner starting by having each convolutional position as a node then pooling the graph based on the edge structure. Different affinity functions based on the node features are used to establish the edges by thresholding. The nodes at the current layer are clustered using a Label Propagation algorithm forming the nodes of the next layer. The features are pooled by taking the first two moments of multiple subsets of each node support and doing a graph convolution. Two decoders, without learnable parameters, are used to render the output, while the node attributes are having the same meaning as the output.

Strengths: An overall good method with sound design choices. Using graph representation in order to obtain object-centric and hierarchical representations is a good direction. The method achieves good results on the proposed datasets, surpassing powerful baselines. The efficiency of the model is a big plus. The model needs up to two orders of magnitude fewer training epochs. The design of the affinity functions in a principled manner is appreciated. The ablation study is appreciated.

Weaknesses: The main weak point of the paper is the presentation. It is very hard to follow, at the same time being very dense but missing key details in the main text and relying too much on the appendix. One possible reason could be over-formalisation. For example, maybe the paragraph 176-184 could be shortened. Details should be left to the supplemental, but the main text should have the main ideas presented in a more clear manner. There should be more details in some parts. More explanations regarding the training should be given. When are the affinity functions optimised as opposed to the main model learned by the QSR and QTR losses? Are they jointly trained? Does the gradient of the affinity function propagate back through the node features into the ConvRNN? It would have been useful to also experiment on datasets already used in the literature such as CLEVR6 or Multi-dSprites. Even though they could be easier, a discussion on the difficulty of solving them compared to the proposed datasets should be useful. ====================================================== Post Rebuttal: ====================================================== I thank the authors for their rebuttal. My main issue with the paper was the clarity of the presentation and the authors seem keen on improving this aspect. I also appreciate the experiments and analysis on the other datasets used in the literature, the zero-shot transfer being particularly interesting. The changes and additions from the rebuttal would improve the paper and thus I will increase my score.

Correctness: The claims of the model are sound and the validation is correct.

Clarity: The paper needs rewrites for clarity reasons. The current form of the paper is detrimental for presenting its contributions. It should explain key parts in a succinct and clear manner.

Relation to Prior Work: More discussion relating to current work in graph representations with GNNs should be added.

Reproducibility: Yes

Additional Feedback: Because the node attributes are having the same meaning as the output, the model is more interpretable and it is easy to manipulate. A discussion could be made for the representation power of the output. How do they compare to the representations obtained when a decoder has learnable parameters? Could such representation be transferred or be used in a reasoning task? Does the model obtain a representation that can be transferred and could generalise to other tasks? Although easily fixable, there are multiple times where standard methods or concepts are not cited. For example VAE or graph convolution.


Review 2

Summary and Contributions: The paper presents physical scene graphs, a self-supervised visual intuitive physics method that automatically reasons about the objects in a scene via incremental segmentation and using motion cues. Contributions include a learnable graph pooling operator, graph vectorization for summarization, graph rendering, an interesting way of object tracking (via tracking edges) and its use for self-supervision, proof of the need non-binary edge affinities and a camera projection loss.

Strengths: - A massive amount of work and contributions presented in minimal amount of space. Super-useful contributions that inspire future work in different visual domains involving GNNs. - Excellent evaluation rigor. - Tests on real data - Good analysis of errors and mistakes in output (L277-L279) - Good actual description of system in supplementary

Weaknesses: - It would be useful to learn how many graph layers are needed to accomplish this general task, e.g. the minimum required value of L without significant loss of performance. Maybe I missed it, but seems to be a valuable piece of information in this dawn of GNNs. - The need for the 6D input in L141 seems to have been not ablated? - Real data only indoors? - (Would be way easier to read if it wouldn’t be limited to 8 pages, 60% of the paper seems to be in the supplementary...suboptimal choice of venue?)

Correctness: Yes, plenty of ablations, mix of synthetic and real datasets and comparison to previous methods make it great.

Clarity: Yes, albeit very very compact.

Relation to Prior Work: Yes, excellently.

Reproducibility: Yes

Additional Feedback: L37: Whilst I might agree with the overall statement about limitations of works similar to [2] and [13], I’d argue that IODINE works with a changing number of objects, even if that number is fixed per scene. There are other methods that do learn from videos in an unsupervised way, e.g. OP3 mentioned later in the paper (L261), and also “Taking Visual Motion Prediction To New Heightfields, Sébastien Ehrhardt, Aron Monszpart, Niloy Mitra, Andrea Vedaldi, ACCV 2018”. I think L37 can be reformulated to better assess the state of science today, whilst still supporting the need for the paper at hand. L258: Self-supervision gets mentioned in L258 together with depth and normal maps being available in the datasets, whilst L266 says only RGB was used as input. It would be useful to clarify what role depth and normals have in the sentence in L258 with relation to self-supervision.


Review 3

Summary and Contributions: I have read the rebuttal which solved some further comments and maintain my strong support for this paper. This work introduces the idea of physical scence graphs (PSGs) to represent arrangement of objects in realistic scenes as well as their hierarchical decomposition into parts. The authors define PSGs and propose and network architecture and a numer of graph operations that extract PSG from real world images. The proposed model outperforms current SOTA for scene segmentation and allows for semantic manipulation of the learned representations.

Strengths: The paper is very clear and well written. The authors propose a novel approach towards scene understanding that is computationally and algorithmically inspired by human cognition. The theoretical exposition as well as the empirical evaluation are extremely thorough. This is an important contribution to the NeurIPS community because it propose a complex new framwork for physical scene understanding along with an empirical demonstration of how this kind of model can be trained and how it improves performance on standard benchmarks.

Weaknesses: The paper is extremely dense and requires considerable background in multiple fields. The exposition of the background and methods takes up more than half of the paper, introducing a large number of design choices, algorithms and inductive biases. It is difficult to understand the significance of each of these specifications, their motivations and how they are implemented in code (e.g. I had to read the whole supp. material to find out that it is implemented in Tensorflow and trained end-to-end). It might be more inclusive for the general ML reader to make the first half of the paper more high-level, introducing the concepts and rough hints at how this can be engineered/implemented. But then accompanied by a long and well structured supplementary text that gives a tutorial in the novel concepts proposed.

Correctness: In disentanglement research randomness across model seeds has considerable effects. Here, in Supp. 320-321 the authors state that performance did not vary much across seeds. Could you expand this claim or show some results to support this? Why is the ConvRNN unrolled 3 times? That means there is an equivalent feedforward model with 15 layers. I understand that feedback is biologically inspired, but is it really necessary here since the same computation could be computed with a deeper feedforward architecture? Did the authors do any benchmarking how long training and inference take in the proposed model? Can they comment on efficient implementations on GPU that allow for varying numbers of nodes in each PSG?

Clarity: The paper is extremely well written. While I suggest that the methodological exposition can be made more concise and readable, I hope that the style and clarity of presented thoughts will remain at this exceptionally high level.

Relation to Prior Work: I would like to see how this work relates to Kulkarni at al.'s 2015 inverse graphics models, as well as the various capsule networks proposed by Hinton et al. Moreover, it would be nice to see how this work relates to recent advances in the learning of disentangled representations (e.g. Locatello et al. 2018), especially given the manipulation experiments in figure 4.

Reproducibility: Yes

Additional Feedback: Main: Figure 1 is much too small. I appreciate the attempt at a simple cartoon-style illustration of the whole framework, but this is too dense and colourful to be informative. 116: I am not familiar/did not find explanations of the big circle notation, can you explain? Supplementary: 31: shown +to 163: where is Delta(v) defined? 340: the -> they


Review 4

Summary and Contributions: This paper proposes Physical Scene Graphs to represent scenes hierarchically and PSGNet that learns to estimate PSGs from visual inputs in a self-supervised manner. Experiments show the validity of proposed methods and generalization ability to real-world images. Detailed ablation studies are conducted to illustrate the importance of each component of the PSGNet architecture.

Strengths: The hierarchical nodes of PSG can locate onto pixels from object subparts to object groupings, which is a natural representation for scenes. The proposed self-supervise learning of such hierarchical representation and its generalization ability to real-world image are novel and experiments are sufficient.

Weaknesses: Some concerns: 1. Previous works on scene graph will model the relations between objects like support relation, occlusion or spatial order. Although hierarchical, PSG only has "edges that represent within-object bonds that hold object parts together", which limits the expressiveness of graph structure. 2. The affinity functions inspired from 4 perceptual grouping principles will encourage the grouping of similarities from many aspects. However it may fail to capture the details of texture in images especially in real-world images, as depicted in Figure 2.

Correctness: With node attributes representing features object position, surface normals, shape and texture properties, making the physical part of scene graph weak. The render and compare methodology is typical in self-supervised/unsupervised scene representation learning.

Clarity: The PSGNet architecture section has too many distinct parts and techniques that need to be addressed, especially the graph construction part. It might be a better idea to further highlight the key points. The experiments are clear and ablation study is well discussed.

Relation to Prior Work: Yes.

Reproducibility: Yes

Additional Feedback: -----Post Rebuttal------ The authors well addressed my concerns and the rebuttal makes the work more clear. Hope to see the corresponding revised version.

[Author Response · NeurIPS 2020]

We thank the four reviewers (**R1**-**R4**) for their constructive feedback and largely positive comments.

**Improving readability.** Several reviewers found the paper clear and "exceptionally well written" (**R3**), but they also
noted the density of the Methods and (esp. **R1**) felt it was overly formal. We agree that this section should better
get across the key ideas, so we will use the extra page allowed in revision to provide higher-level explanations and
background for each concept we introduce. To further improve readability, we will also make the following changes:

★ Lines 108-120, the description of PSGs, will be made less formal. The key point is that PSGs are hierarchical graphs
with additional structures that connect to visual input. Within-level edges are meant to represent physical connections;
parent-to-child edges are meant to represent part-whole relationships; attribute vectors are meant to encode physical
properties of elements of the visual input; and spatial registrations map each vertex in the graph to a subset of pixels in
the input movie. Because a PSG is hierarchical, the registrations form a hierarchical segmentation of a scene.

★ Lines 121-133 should better address **R1**'s questions about model training. In particular, PSGNets have two types
of losses that are *trained jointly*: (1) *rendering losses*, which train the parameters of Graph Vectorization and Feature
Extractor modules by backpropagation, because the rendered feature maps are differentiable functions of predicted
attribute vectors and image features, respectively; and (2) *perceptual grouping* losses, which train the affinity functions
in Graph Pooling modules. Rendering gradients *cannot* flow back into the affinity functions because Label Propagation
is not differentiable; perceptual grouping gradients *can* flow into the other modules. To clarify L258 (per **R2**), we
will emphasize that depths and normals can *supervise* rendering losses (1) but are never used as input. We describe
further experiments without any depths and normals supervision below. Perceptual grouping losses (2) never receive
supervision. We will implement **R3**'s idea to make Fig. 1 larger by splitting it into Architecture and Training figures.

★ We can simplify the rest of the Methods as five shorter subsections: (1) ConvRNN Feature Extraction: explain that a
ConvRNN, unlike a deep CNN, generates features *from a single layer* with different useful properties after each recurrent
pass (see Supplement); (2) Graph Pooling: emphasize ideas of affinity functions and explain why Label Propagation
prevents rendering gradients from training this module; (3) Graph Vectorization: Move details of aggregation to
Supplement (e.g. L176-184, per **R1**), emphasize taking statistics over segment interiors and boundaries (to encode
shape information) and predicting attributes *via* Graph Convolution [Kipf & Welling, *ICLR* 2017]; (4) Rendering:
Combine the QTR, QSR, and Losses sections to explain why parameter-free decoding forces node attributes to encode
scene properties *explicitly*; discuss (per **R1**) why this leads to interpretable representations, as illustrated by graph
editing (Figure 4); (5) Perceptual Grouping: explain why pairwise inductive biases make sense for grouping and how
$\beta$-VAEs [Higgens *et al.*, *ICLR* 2016] can naturally encode the idea of node co-occurrence and motion-in-concert.

**Data to address Reviewer 1. R1** makes an excellent suggestion to compare the MultiDSprites (MDS) and CLEVR6
datasets with our custom datasets. We thus built PSGNetS-RGB, a compact version of the original model that does not
use depths/normals supervision. Trained/tested on the Playroom dataset, PSGNetS-RGB achieved scores of (**recall**
0.59, **mIoU** 0.55, **boundary F** 0.57), which are (as expected) somewhat, though not dramatically, lower than the
original PSGNetS scores. Without further hyperparameter tuning, PSGNetS-RGB trained/tested on MDS achieved
(**r**0.80, **m**0.70, **b**0.72); trained/tested on CLEVR6, it achieved (**r**0.73, **m**0.63, **b**0.67). The higher scores on MDS and
CLEVR6 suggest that the Playroom dataset is indeed harder to segment than those used in the literature. Moreover, in
running these experiments we found something else worth reporting: the Playroom-trained PSGNetS-RGB achieved
*zero-shot transfer performance* of (**r**0.75, **m**0.67, **b**0.71) on MDS and (**r**0.70, **m**0.60, **b**0.66) on CLEVR6, nearly as
high as the within-dataset scores. This further indicates that PSGNet inductive biases lead to fairly general object-
centric representation learning. We will include these points in the revision, along with MDS/CLEVR6 scores for our
implementations of the baseline models (which take longer to train than the allotted Author Response period.)

**Data to address Reviewers 2 & 3.** The Playroom-trained PSGNetS-RGB model above ablates the three $\delta x$ input
channels, as per **R2**. To measure the effect of ablation, we restored these channels to create PSGNetM-RGB and trained
five models with different random seeds. These achieved scores (mean +/- stdev) of (**r**0.65 +/ 0.01, **m**0.59 +/- 0.01,
**b**0.60 +/- 0.01), substantially higher than PSGNetS-RGB. This indicates that the $\delta x$ input channels are useful for
segmentation and supports our claim of low performance variability, noted by **R3**. Each model took ~24 hours to train
on one Titan Xp GPU; inference takes ~200 ms/image. PSGs are implemented on a GPU by choosing a *maximum* node
number per level, then masking out nodes that have no incoming child-to-parent edges (& thus represent nothing.)

**Response to Reviewer 4. R4** raised two concerns about (1) the types of edges in the PSG and (2) the inability to group
by real-world textures. We want to clarify that these are *not* inherent limitations of our approach, but rather choices
we made to limit the scope of the submission. As to (1), PSGs can naturally incorporate additional edge types, for
example to represent temporary occlusions, collisions, or support relationships between objects. We use these edge
types in ongoing work to predict physical dynamics with PSGs. As to (2), by self-supervising rendered feature maps on
higher-order statistics of the input image, such as the covariance of RGB pixels within each node segment, attribute
vectors can be trained to explictly encode texture. Perceptual grouping can take advantage of these texture components.

[Meta-Review · NeurIPS 2020]

All reviewers agree that this is a strong paper with solid experiments. My recommendation is accept. Please take the reviewers' comments into account in preparing the final version of the paper.